# Metal Chelation Therapy and Parkinson’s Disease: A Critical Review on the Thermodynamics of Complex Formation between Relevant Metal Ions and Promising or Established Drugs

**DOI:** 10.3390/biom9070269

**Published:** 2019-07-09

**Authors:** Marianna Tosato, Valerio Di Marco

**Affiliations:** Analytical Chemistry Research Group, Department of Chemical Sciences, University of Padova, via Marzolo 1, 35131 Padova, Italy

**Keywords:** Parkinson, neurodegeneration, chelation therapy, Alzheimer’s disease, conservative chelation, Amyotrophic Lateral Sclerosis

## Abstract

The present review reports a list of approximately 800 compounds which have been used, tested or proposed for Parkinson’s disease (PD) therapy in the year range 2014–2019 (April): name(s), chemical structure and references are given. Among these compounds, approximately 250 have possible or established metal-chelating properties towards Cu(II), Cu(I), Fe(III), Fe(II), Mn(II), and Zn(II), which are considered to be involved in metal dyshomeostasis during PD. Speciation information regarding the complexes formed by these ions and the 250 compounds has been collected or, if not experimentally available, has been estimated from similar molecules. Stoichiometries and stability constants of the complexes have been reported; values of the cologarithm of the concentration of free metal ion at equilibrium (pM), and of the dissociation constant *K*_d_ (both computed at pH = 7.4 and at total metal and ligand concentrations of 10^−6^ and 10^−5^ mol/L, respectively), charge and stoichiometry of the most abundant metal–ligand complexes existing at physiological conditions, have been obtained. A rigorous definition of the reported amounts is given, the possible usefulness of this data is described, and the need to characterize the metal–ligand speciation of PD drugs is underlined.

## 1. Introduction

Parkinson’s disease (PD) is a common neurodegenerative disorder (ND) [1]. It is characterized by neuronal cell loss in the substantia nigra (SN), which leads to a progressive central nervous system dysfunction. Symptoms include motor abnormalities like tremors, movement and balance issues, and non-motor problems like difficulty in swallowing and speaking, depression, cognitive impairment, and dementia. Although PD by itself is not a fatal disease, people may die from causes related to it.

Age is the most relevant risk factor: approximately 2% of people over the age of 60 years, and 3% of those at age over 80 years, suffer from PD [2]. Due to the average population aging, the occurrence of PD and of other NDs like Alzheimer’s disease and Amyotrophic Lateral Sclerosis is continuously increasing. The number of persons suffering from PD is expected to reach a value of approximately 9 million in 2030 [3] and up to more than 17 million in 2040 [4]. NDs are considered the leading source of disability around the world, and the fastest growing of these disorders is PD [5]. Although non-infectious, PD exhibits many of the characteristics of a pandemic, and it is experiencing exponential growth worldwide [4,6,7]. Together with the average population age increasing, other factors will likely contribute to increase the incidence of PD over current forecasts. In particular, reducing smoking rates in some countries may lead to a higher incidence of PD, because many studies have found that the risk of this disease is decreased among smokers by approximately 40% [8]. Also, pollutants by-produced in industrialized countries may contribute to the rising rates of PD, because specific pesticides, solvents, and heavy metals have been linked to this disease [9]. Actually, countries that underwent the most rapid industrialization have seen the greatest increase in the rates of PD [5].

PD can be due to genetic factors, and it has been found that race/ethnicity can affect the incidence of PD in the order Hispanics > non-Hispanic Whites > Asians > Blacks [10]. However, genetics appears to justify only a small amount (approximately 5–10%) of all PD cases: it is therefore possible to suggest an important role of external factors, like behavioural and environmental [2,9,11].

Existing drugs for PD provide only the relief of some symptoms, and there are no disease-modifying therapies demonstrated to slow or to stop the ongoing neurodegenerative process. In the search for such therapies, however, the primary issue lies in the multifactorial nature of PD. The main neuropathological hallmark of PD is proteopathy, as the formation and deposition of protein aggregates is generally observed in PD brains. The most typical deposits, called Lewy bodies, are due to α-synuclein (α-syn), an abundant neurological protein with yet unclear physiological functions. Other features characterizing PD brains are mitochondrial dysfunction, oxidative stress, defects in energy metabolism, aberrant axonal transport, and metal ion dyshomeostasis [12]. All the pathways operating in PD appear to be strictly inter-related, so that both the study of the molecular causes of the disease, and the search for an efficient therapy, cannot be limited to a specific pathway: a multi-targeted approach is suggested [1,12]. Much scientific effort in recent years is devoted to the comprehension of each pathological mechanism operating in PD, with the aim to understand and rationalize the biochemical processes occurring during the pathology. This review focuses on metal dyshomeostasis and on the therapy that is aimed to target this pathological process, i.e., metal chelation therapy.

## 2. Parkinson’s Disease and Metal Ions

Almost one hundred years ago, Lhermitte et al. [13] discovered that the brains of people who died from Parkinsonism, a form of dementia with similar symptoms as those of PD [14], contained a significantly larger amount of iron (Fe) than the corresponding brains of controls. Since that work, several studies have confirmed the abnormally high Fe content also in the brains of PD patients [15,16,17,18,19]. Conversely, other studies could not detect an overload of Fe in PD brains [20,21]. Recently, several studies have attempted to determine Fe brain levels in living patients with PD ([22,23] and references therein). Most but not all studies indicated larger Fe levels in the SN of patients with PD compared to control subjects, whereas no Fe excess was observed in many other brain parts, thus suggesting that a Fe dyshomeostasis occurs in PD brain, especially in the SN [23,24]. Bush et al. found that the reported Fe accumulation is contributed to by a disturbance in Fe export. This was explained by a significant reduction of the specific activity (but not of the levels) of ceruloplasmin in the SN of PD patients [25]. Ceruloplasmin is a multicopper ferroxidase protein facilitating cellular Fe export [26]. The same authors [27] observed a decrease in the PD SN of the soluble levels of another protein, tau, which can lower neuronal Fe levels by promoting the presentation of the amyloid protein precursor to the neuronal surface, where it favours the efflux of Fe [28]. It has also been suggested that PD-induced Fe accumulation is due to a dark pigment contained in SN, neuromelanin, that is able to bind Fe and that may act as a protection against Fe by binding and storing its excessive labile content [29].

A number of studies have also shown alterations in the copper (Cu) concentrations in the brain of post-mortem PD patients compared to non-PD controls [11,30], suggesting that metal dyshomeostasis in PD brains also regards Cu [31]. However, while Fe appeared to be systematically overloaded, Cu was significantly reduced in the degenerating regions of PD brains [30,32,33,34]. Other metal ions have been monitored in PD brains, but less definite results were reported. Conflicting data were reported for zinc (Zn) [35,36] and Genoud et al. recently evidenced no differences in Zn levels between experimental groups [30]. Parkinsonism is reported to rapidly develop in patients subjected to the exposure of high levels of manganese (Mn) [11,37], and a role in Parkinsonism onset has also been suggested for Cu [38]. However, no changes have been detected in the Mn level of post-mortem PD brains with respect to non-PD samples [30], and, as seen, even reduced levels were detected for Cu. Other elements which have been occasionally linked with PD have been aluminium (Al), arsenic (As), bismuth (Bi), cadmium (Cd), mercury (Hg), lead (Pb), thallium (Tl), and titanium (Ti) [2,11]. Bjørklund et al. [11] reviewed the works in which the exposure to metal ions was shown to correlate with the onset of PD and/or of Parkinsonism.

These results have prompted researchers to clarify the role of each metal ion in PD. In literature, the most studied metal ions have been Fe, Cu, Mn and Zn, whereas papers regarding other elements were much fewer. The huge work performed on this matter has been reviewed in detail [11,31,32,39,40,41], and it regarded the molecular mechanisms and biological aspects of these elements in the brains of controls and of PD patients.

Essential metal ions like Cu, Fe, Mn, and Zn are known to be involved in a large number of biochemical processes in the human brain [41,42], where they exert a structural (e.g., stabilizing configurations of macromolecules) or a functional role (e.g., being the active site of metalloenzymes). Both Cu and Fe can exist in vivo under two oxidation states, Cu(II) and Cu(I), Fe(III) and Fe(II), to allow biological systems activating and using O_2_ for energy purposes. Reactions activating O_2_, if not tightly regulated, can cause oxidative stress, so that healthy biological systems contain suitable antioxidants and very little exchangeable Cu and Fe ions. This metal ion fraction is also called “labile” or “free” ion [43], and it is thought to be the main contributor of metal-induced oxidative stress [24,44,45]. Metal ions in the labile pool can be loosely bound to peptides, carboxylates and phosphates as compounds with low mass, while some might exist as hydrated free ions. In healthy mammalian cells, the labile Fe concentration is less than 1 µmol/L, and less than 5% of total Fe [46]. The labile Cu and Fe fractions exert their toxicity by generating reactive oxygen species (ROS) via the Fenton and the Haber–Weiss reactions, both related to the presence of the Fe(III)/Fe(II) or of the Cu(II)/Cu(I) redox couple. The Fenton reaction for Fe is:Fe(II) + H_2_O_2_→ Fe(III) + OH• + OH^–^

Fe(III) + H_2_O_2_→ Fe(II) + OOH• + H^+^(1)

In PD brains, a dysregulation occurs between the production of OH• and OOH• (and of other ROS) and their removal, thus resulting in cellular damage through the oxidation of lipids, proteins, and DNA. The levels of glutathione, one of the most important antioxidants in human brain, were reported to be significantly decreased in the SN of PD patients compared with those of healthy subjects [47]. A dyshomeostasis of Fe or Cu can therefore have a significant impact on ROS regulation. Also, high labile Mn levels have been reported to increase oxidative stress [42]. Labile Zn^2+^ is more abundant in healthy brain cells, as it is released by neural activity at many central excitatory synapses [48], but still this metal ion was related to oxidative stress [49].

Labile metal ion pools can also undergo a pathogenic relation with α-syn. Being an unfolded protein, α-syn can easily switch in a number of conformational states in response to changes in environmental conditions [50,51]. Temperature changes, presence of pro-oxidative conditions [52,53] and of several metal ions can promote the formation of dimers and other polymeric forms of this protein [54,55]. The misfolding of α-syn is thought to be the most important factor driving the formation of Lewy bodies in PD, and, in turn, toxic forms of aggregated α-syn are released from neurons, and then spread between cells in a prion-like manner [32]. It was shown that part of RNA structure posttranscriptionally regulates α-syn production in response to cellular Fe and redox events [56,57], so that the overexpression of α-syn promotes the neuronal accumulation of Fe. Fe can promote the aggregation of this protein, and post-translational modifications of α-syn have also been found to regulate Fe transport [58]. It was found that α-syn can inhibit the lysosome-mediated degradation of ferritin (a Fe storing protein), resulting in the intracellular build-up of ferritin and consequently of Fe [59]. Also, the direct interaction between metal ions and α-syn in neurons, with the formation of metal–protein complexes, can be of primary importance to justify the protein unfolding and eventually its aggregation. Furthermore, the complexes themselves may be cytotoxic, as, e.g., it was reported to be Cu^2+^/α-syn [60]. The properties of the complex formation between metal ions and α-syn have been reviewed by several authors (e.g., [12,32,60,61]). As regards the binding moiety, it is known that metal ions can bind to high-affinity N-terminal (containing residues 1–60) and to lower affinity C-terminal sites of α-syn (from 96 to 140 amino acid residues). Studies have been performed to evaluate the stability of the metal–α-syn interactions: information available in the literature, given as dissociation constants (*K*_d_, see below) of the complexes formed between metal ions and α-syn, is resumed in Table 1.

A recently discovered cell death pathway, called ferroptosis or Fe-dependent cell death, has provided further impetus to the “Fe hypothesis” of PD, Alzheimer’s disease, and Amyotrophic lateral sclerosis [39,62,63]. Fe still has an unclear role in ferroptosis, but it has been shown that Fe chelation is beneficial in preventing this cellular damage, which is also characterized by increased levels of lipid hydroperoxides and by a depletion of the important antioxidant glutathione. PD has been linked to ferroptosis because literature studies generally indicate that ferroptosis inhibitors may be effective in PD, too [39]. For example, chelation therapy can be beneficial in PD (see below), and prodrugs such as *N*-acetylcysteine, which enhances glutathione levels in brain, exert partial protection against PD neurodegeneration. 

Many mechanisms have been considered and reviewed for the damage induced by Cu [44], Fe [49,64,65,66], Mn [67] and Zn [49] under PD conditions; in several cases, oxidative damage, metal dyshomeostasis and α-syn aggregation have been demonstrated to be strictly related to each other.

Despite the availability of many useful results, the molecular pathways describing the association between metals and PD onset are still ill defined. It remains controversial whether the dyshomeostasis of Cu, Fe, Mn, Zn, and possibly of other elements, is the primary cause or secondary consequence of PD as well as of other neurological diseases such as Alzheimer’s disease, multiple system atrophy, dementia with Lewy bodies, amyotrophic lateral sclerosis, Huntington’s disease, frontotemporal dementia, corticobasal degeneration, and progressive supranuclear palsy. The possibility that metal dyshomeostasis is just a secondary consequence of other independent molecular paths is supported by considering that the timing for PD onset is much slower (many years) than for Parkinsonism, which in turn is very rapidly induced if, e.g., Mn or other external toxins are administered to animal models. This suggests that endogenous neurotoxins, rather than exogenous ones, are responsible for the extremely slow neurodegeneration observed in PD. In their very recent review, Ndayisaba et al. tried to answer whether neurodegeneration is caused (or co-caused) by Fe dyshomeostasis, whether the latter contributes accelerating the pathological effects due to nerve cell death and to release of intracellular components, or whether neurodegeneration is simply not related to Fe accumulation [68]. The authors were not able to give a definite answer, but they observed that Fe dyshomeostasis occurs already at early PD onset, and that Fe should at least contribute to many aspects of neurodegeneration, in a way such that Fe might be proposed as a biomarker to detect for preclinical stages of NDs. Similarly, prudent conclusions have been drawn in another very recent review by Chen et al. [42], where the authors found that it was unclear whether Fe and Mn are primary or secondary causes of neurodegeneration, as they found that neurodegeneration cannot be reversed if metal overload is removed. The question about the primary or secondary role of Cu and Zn was raised by Barnham and Bush [41] in their review. They concluded that PD, Alzheimer’s disease, Huntington’s disease and amyotrophic lateral sclerosis are not caused by a simple overload of these metals, although the possibility that exposure may alter disease risk was not excluded. The authors found a number of possible molecular pathways induced by altered Cu and Zn levels in PD brains, which can at least contribute to the disease progression. The authors concluded that “the targets of metalloneurobiology are rich with pharmacological opportunities” [41].

## 3. Metal Chelation Therapy in Parkinson’s Disease

Metal chelation therapy (MCT) was proposed more than 50 years ago for the therapy of pathologies produced in the body by an overload of a metal. Metal chelation therapy involves the use of a chelating agent (CA), i.e., a molecule which forms stable coordination complexes with the target metal ion. Once administered to the patient, the CA acts as a scavenger removing the metal from its stores and favouring its decorporation from the body [69]. An efficient CA should be orally active and have a low cost, and both the ligand and the complexes formed in vivo should possess suitable hydrophilicity/hydrophobicity, and no redox activity [70]. In particular, the CA affinity towards the overloaded metal ion should be as high as possible. Last but not least, CAs and their metal complexes should display no toxicity and no or negligible side effects, but these properties are still only partially verified for established CAs. For example, the common Fe and Al chelator Desferrioxamine (also known as Desferal, DFO or Deferoxamine) is reported to cause a number of severe side effects [71], among which heart diseases [72] and retinopathy [73] appear to be the most important ones. Adverse effects were also reported for the other two established Fe chelators, Deferiprone and Deferasirox. According to Fisher et al. [74], side effects increased in patients treated with Deferiprone compared with Desferrioxamine. Kontoghiorghes et al. [71,75] reported a number of fatal renal, liver, and bone marrow failures due to Deferasirox. Another common CA used for Cu overload, D-Penicillamine, causes neuropsychiatric or hepatic complications in up to one-third of patients [76]. Toxic effects have also been observed with other CAs (2,3-dimercaptopropanol, meso-2,3-dimercaptosuccinic acid, 2,3-dimercaptopropane-1-sulfonic acid, EDTA calcium or sodium salts) used in the therapy for the overload of As, Au, Hg, Pb [77].

Despite these toxicity reports, the occurrence of a metal ion dyshomeostasis in PD has suggested to also employ MCT for the therapy of this disease and of other NDs such as corticobasal degeneration, the Westfal variant of Huntington’s disease, Alzheimer’s disease, Friedreich’s ataxia, pantothenate kinase-associated neurodegeneration, and other neuropathologies associated with brain metal overload [12,24,78,79]. In these cases, MCT was also referred to as “metal targeting”, “metal attenuating” or “metal protein attenuating” [41,80,81], in order to underline the differences occurring when MCT is employed in NDs instead of in metal overloads. Poujois et al. [82] have further improved this chelation strategy, and they called it “conservative chelation”. For the design of metal-based therapeutic strategies in NDs, the complete removal of metals from affected tissues is not the desired mechanism of action of these drugs. The terms “targeting”, “attenuating” and “conservative” evidence that the CA should remove only labile essential metal ions, which are considered to be not functionally required. Essential metal ions are aimed to be removed from the biological targets where they might be harmful, in particular, to avoid the α-syn complex formation and ROS-generating redox reactions, but they should also be allowed acting their normal physiological functions, as for example in metallo-enzymes, thus preventing severe side effects. Another important feature of a conservative CA is that the labile metal pool should be redeployed to cell acceptors or transport proteins (e.g., transferrin for Fe) [82]. This mode of action is expected to correct aberrant metal distribution, minimising systemic loss of chelated metal, thus avoiding the CA to cause metal-deficiency anemia and interfere with metal-dependent mechanisms essential for normal physiological functions. Conservative chelation, instead of a more aggressive metal removal, appeas to be particularly suitable in PD if Fe is the target, as patients suffering from this ND are mainly elderly people who are often on the border or already with Fe deficiency. Nevertheless, the unspecific removal of any essential trace metals may lead to harmful adverse effects to people of all ages, and metal deficiency can be regarding not only targeted but also untargeted metal ions. Excessive removal of Cu and Zn has been often reported for β-thalassemic patients undergoing MCT with Fe-chelators such as Deferiprone, and especially Desferrioxamine and Deferasirox [83,84]. Cu and Zn anemia, in turn, can lead to delay in growth and development, immunodeficiency, and abnormal hematopoiesis [84]. Metal redistribution, rather than metal removal, is therefore the goal in PD [24,85]. To allow a conservative chelation, a CA to be employed in PD should form moderate but not too strong complexes with the target metal ion. As an additional property, the drug should be able to pass the blood–brain barrier [78]. The ability of a CA to pass this barrier can be improved by the prochelator strategy, which for PD and also for other pathologies has been extensively reviewed by Oliveri and Vecchio [86].

In PD, MCT was proposed to target dysregulated essential metal ions [85], mainly the labile pool of Fe and Cu [24,78] but also of Zn [81] and, more rarely, of Mn [40], rather than for the decorporation of total toxic ions such as Al, Hg, Pb, etc. MCT has been tested in a number of translational studies on cell lines or on animal PD models, where Parkinsonism was induced by the administration of OHDA (6-hydroxydopamine) or MPTP (1-methyl-4-phenyl-1,2,3,6-tetrahydropyridine). The Fe chelator Desferrioxamine was reported to reduce iron-induced oxidative stress in SK-N-SH cell line and dopaminergic cells aggregation [87], and its intranasal administration significantly improved PD symptoms in MPTP-treated mice [88]. The other Fe chelator Deferiprone, differently than Desferrioxamine, is orally active and is better able to cross the blood–brain barrier [89]. Deferiprone demonstrated to be efficacious in MPTP and OHDA-induced animal models of PD [90]. Other CAs tested in cell lines or in animal models for the PD therapy have been Clioquinol [91], VK-28 [92,93], M30 [94], PBT2 [95], Q1, Q4 [96] and several other compounds, as reviewed recently by Singh et al. [89]. 

The first clinical evidence about the efficacy of a conservative Fe chelation regimen for human PD was given by Devos et al. [90], who orally administered Deferiprone to PD patients for 12 months. The Fe deposits in the SN were significantly reduced, and the Unified Parkinson’s Disease Rating Scale motor indicators of disease progression were significantly improved. However, when the treatment was suspended, Fe started to accumulate again, suggesting a reversal to the pathological state. Deferiprone, differently than other well-known Fe chelators such as Desferrioxamine and Desferal, has the important feature to rescue transfusional hemosiderosis in the hearts of β-thalassemia patients without inducing significant anemia, largely attributable to the redeployment of captured Fe to extracellular iron-free transferrin and subsequent distribution [97], thus allowing this CA to be employed for a conservative chelation strategy. Devos et al. [90] reported that none of the Deferiprone-treated PD patients developed new neurological signs, and no level changes (of Fe and of other transition metals) were detected in brain parts not involved in PD. The conservative chelation strategy used by the authors prevented side effects typically due to Fe-deficiency anemia in the brain such as the restless legs syndrome [98]. Other Fe chelators forming stronger complexes with the targeted metal ion could have caused such unwanted side effects as they would likely also remove the non-labile part of Fe or of other metal ions [90]. The authors concluded that Deferiprone can represent a paradigm for conservative Fe chelation. The encouraging results obtained for this CA prompted the development of other clinical trials with Deferiprone. A search in https://clinicaltrials.gov indicated four ongoing or finished tests of this molecule for the treatment of PD, as also recently reported by Nuñez and Chana-Cuevas [79]. 

To the best of our knowledge, no other CA is still being subjected to clinical trials. However, other than Deferiprone, there is a number of molecules which have been or are being considered good therapeutic candidates for PD therapy [24,79,85,89]. Nowadays, due to the multi-faced nature of PD, the proposed strategy for MCT requires the use of multifunctional molecules able not only to bind metal ions thus controlling metal dyshomeostasis, but also to counterbalance other toxic pathways in PD. Multifunctional molecules for PD have recently been reviewed by Savelieff et al. [12].

All previously cited reviews list the names and sometimes the chemical structures of CAs used as or tested for PD therapy. However, these reviews do not report which complexes can form between relevant metal ions and CAs (metal–ligand stoichiometries) and how stable they are (metal–ligand stability constants), i.e., they lack speciation information. Some lists of metal–ligand stoichiometries and of stability constants (given as log*β*, see below) for promising PD drugs have been reported, for example by Gumienna-Kontecka et al. [24], Kasprzak et al. [99], and Prachayasittikul et al. [100]. The Gumienna-Kontecka log*β* list was however limited to very few compounds and to the complexes formed with Fe(III), the Kasprzak and Prachayasittikul lists report only the complex formation of flavonoids and 8-hydroxyquinolines, respectively. Many of the reported lists also lack the ligand acidity constants, which (see below) are necessary to achieve a complete speciation picture.

The knowledge of metal–ligand speciation can allow modeling the CA activity in vitro or in vivo, i.e., to perform calculations (e.g., [101]) describing the distribution of the metal ion of interest at any conditions. Some examples of information which can be gathered from speciation calculations in the frame of MCT will be briefly given below. We think, and it has been stated (see, e.g., [101,102]), that knowledge about metal–ligand speciation should be easily available to researchers interested in the study of pathologies involving metal ions, such as PD. As regards CAs, the aforementioned knowledge should at first regard all essential metal ions (in all their possible oxidation states) undergoing dyshomeostasis in PD. The relevant ions are therefore Cu(II), Cu(I), Fe(III), Fe(II), Mn(II), and Zn(II) (Mn(III) should also be considered in principle, however we decided to exclude it because its content in the body is generally considered to be minimal [103]).

It is also necessary to note that there are several other established or promising PD drugs, not specifically designed for MCT, which can also act as CAs. A typical example is L-Dopa (levodopa), the gold standard in the therapy against PD: its pharmacological activity is aimed to increase the dopamine deficit in vivo, but it can also form stable complexes with several metal ions including the PD relevant ones [104]. These compounds are by themselves multifunctional drugs, and their potential influence on the pathological mechanisms involving metal ions cannot be neglected. In the present review, all drugs displaying metal chelation properties, not only those specifically designed for MCT, will be considered.

A bibliographic search has been performed with the aim to collect all promising and established drugs for the PD therapy. Keywords and boolean logics employed for this bibliographic search are given in the Appendix A. Several reviews appeared in 2014 or later, reporting lists of anti-PD compounds [1,12,105,106,107,108,109,110]. These reviews list mainly molecules in use, or which underwent in vivo, or clinical phase tests against PD; here, we also decided to consider compounds which have just been tested in vitro, or even only proposed, e.g., after an in silico approach, because it is likely that some of these will undergo further tests in the following years. Clearly, if a compound was proposed and/or tested in vitro several years ago, and after not more considered as anti-PD drug, likely it was not suitable for this aim and has been abandoned. Due to the latter consideration, and given the availability of the above listed reviews, we decided to limit our bibliographic search to papers published starting from 2014. 

For each compound, name(s), chemical structure and the reference(s) were collected. The whole list of these substances is reported in Appendix A. This table lists approximately 800 compounds and, to the best of our knowledge, it is the most complete table available to date which reports established or potential anti-PD drugs. Appendix A also contains the compounds listed in all above mentioned reviews of PD drugs [1,12,24,79,85,89,104,105,106,107,108,109,110]. Appendix A does not report natural extracts, e.g., drugs obtained from plants or animals, unless the active components have been identified. This therapeutic approach is extensively considered in the literature: for a recent review see, e.g., [111].

Table 2 represents a subset of Appendix A and it lists all compounds (nearly 250) displaying metal-chelating properties which have been used, tested or only proposed for the therapy against PD.

It is rather easy to predict whether a given compound can form stable complexes with metal ions, and thus whether it in principle can affect metal homeostasis in the brain: the presence of at least two functional groups with metal-binding ability is suggested, and the formed chelation ring should have five or six members. Truly, monodentate ligands and those forming chelating rings of a different size to 5 or 6 can also coordinate metal ions, but the resulting complexes are generally too weak to allow these ligands to affect metal speciation in vivo. Coordinating functional groups can be negatively charged (or partially charged) oxygens such as carboxylates, phenolates, *N*-oxides, as well as nitrogen and sulphur atoms with non-delocalized lone pairs such as amines and thio-derivatives. For example, l-Dopa (chemical structure drawn in Appendix A) has two possible binding sites, one given by the two phenolic oxygens (catechol-like), and the other by the amino-acidic tail (glycine-like). Both binding sites, upon chelation, form a five membered ring with the metal ion. Some simple metal-chelating moieties often contained in CAs are depicted in Figure 1

The formation of stable metal–ligand complexes is more difficult to predict for peptides, because their metal chelation ability is strongly influenced by their spatial configuration. Also, it is not possible to assess the metal–ligand speciation of molecules bearing many chelating functional groups. Molecules of this kind, cited in Appendix A, were not reported in Table 2. For the relevant metal ions (Cu(II), Cu(I), Fe(III), Fe(II), Mn(II), and Zn(II)) and for the compounds listed in Table 2, the metal–ligand speciation will be given.

## 4. The Measurement of the Stability of Metal–Ligand Complexes

While no ambiguities exist for reporting the stoichiometries of the complexes, it is worth describing all possible amounts used in literature to measure the stability of metal–ligand complexes: cumulative stability constants, stepwise stability constants, conditional stability constants, cologarithm of the concentration of free metal ion at equilibrium (pM), association constants, and dissociation constants.

Cumulative (or overall) stability constants are generally indicated with the Greek letter *β*. Each complex forming in solution is characterized by a *β* value. If M is the metal ion, H the proton, and L the ligand, and M*_m_*H*_h_*L*_l_* is the complex formed, *β* is defined as:(2)$$\beta =\frac{\left[M_{m}H_{h}L_{l}\right]}{{\left[M\right]}^{m}{\left[H\right]}^{h}{\left[L\right]}^{l}}$$
where square brackets denote concentrations at equilibrium. The use of concentration amounts instead of activities (i.e., concentrations multiplied by activity coefficients) is generally allowed by maintaining a constant ionic strength during the experimental measurements. The (at least formal) presence of activities in equation (2) justifies why *β* values are commonly indicated without a measuring unit. Cumulative constants are usually given as logarithm (log*β*), and their knowledge is required for performing metal–ligand speciation calculations. Their experimental determination is however complicated as many experimental details have to be considered in order to obtain accurate results [312,313]. In addition, *β* values do not allow to state the effective complex stability, which is also affected by the acid—base properties of metal ion and ligand, by the total metal (*c*_M_) and ligand (*c*_L_) concentrations, and by the pH. In other words, log*β* values by themselves are not informative and cannot be used to compare the stability of complexes formed by different metal ions and ligands.

Stepwise stability constants are generally indicated with the letter *K* and they are written as log*K*. Stepwise constants are more commonly employed if the complexes existing in solution contain one metal ion and one or more ligands (ML*_l_*, with *l* ≥ 1). For example, for the complex ML*_l_*, *K* can be defined as:(3)$$K=\frac{\left[ML_{l}\right]}{\left[ML_{l-1}][L\right]}$$

Stepwise constants are related with cumulative ones, so that the former can be computed from the latter and vice versa— e.g., for the complexes ML and ML_2_, log*β*_ML_ = log*K*_ML_ and log*β*_ML_2__ = log*K*_ML_ + log*K*_ML_2__, respectively.

Conditional (or effective) stability constants may be cumulative or stepwise and are indicated with the apostrophe (*β* ‘ or *K* ‘). For example, the conditional cumulative constant of the complex ML_2_ is defined as:(4)$$\beta \prime =\frac{\left[ML_{2}\right]}{\sum [M\prime ](\sum {\left[L\prime ]\right)}^{2}}$$
where Σ[M′] and Σ[L′] represent the sum of the concentrations of uncomplexed metal ion and uncomplexed ligand at equilibrium, i.e., [M] + [M(OH)] + [M(OH)_2_] + … and [L] + [HL] + [H_2_L] + …, respectively. As Σ[M′] and Σ[L′] depend on pH, also log*β*′ and log*K*′ values depend on pH. These constants represent the effective stability of the given complexes in the presence of acid—base equilibria, so that they can be used to compare the stability of complexes formed by different metal ions and ligands. However, the comparison is possible only if the complexes have the same stoichiometries.

The other three amounts used to measure metal–ligand affinities, i.e., pM, association constants, and dissociation constants, differ from log*β*, log*K*, and from their conditional values, because only one number is given to characterize a solution containing the given metal and ligand. This is particularly useful when many complexes are formed and, in overall, one value resumes their strength.

pM represents the cologarithm of the concentration of free metal ion at equilibrium (pM = –log[M]), and it can be calculated if log *β* or log*K* values are known. The larger is pM, i.e., the lower is [M], the stronger are the complexes; pM can be used to compare the relative strength of the complexes, irrespective of their number and their stoichiometry. As pM depends on pH and also on *c*_M_ and *c*_L_, calculation must be performed under the same conditions: usually the pM value for MCT relevant conditions is computed at pH = 7.4, *c*_M_ = 10^−6^ mol/L and *c*_L_ = 10^−5^ mol/L [24,70,314]. The only (but important) disadvantage of pM is that it can be computed if log*β* or log*K* values are known, i.e., the experimental procedure required to gain a pM value remains complicated.

Association constants, indicated as *K*_a_, are defined like *β* or *K* if only one complex ML forms in solution and no acid–base equilibria coexist:(5)$$K_{a}=\frac{\left[ML\right]}{\left[M][L\right]}$$

If M and/or L undergo acid–base equilibria, *K*_a_ for the complex ML is defined like *K*′ or *β*′. When more complexes of general stoichiometry M*_m_*H*_h_*L*_l_* coexist in solution, more *K*′ or *β*′ are needed, whereas still only one *K*_a_ suffices and is defined as:(6)$$K_{a}=\frac{\sum \left[M_{m}H_{h}L_{l}\right]}{\sum [M\prime ]\sum \left[L\prime \right]}$$
where Σ[M*_m_*H*_h_*L*_l_*] represents the sum of the concentrations of all complexes existing in solution. As *K*_a_ values are not thermodynamically defined, i.e., concentration values are employed instead of activities, they bear a measuring unit, which according to equation (6) is L/mol (or, more commonly, a multiple).

The dissociation constant, indicated as *K*_d_, is the inverse of *K*_a_ (measuring unit of *K*_d_: mol/L or a multiple). For example, if complexes of general stoichiometry M*_m_*H*_h_*L*_l_* coexist in solution, *K*_d_ can be defined as:(7)$$K_{d}=\frac{\sum [M\prime ]\sum \left[L\prime \right]}{\sum \left[M_{m}H_{h}L_{l}\right]}$$

Kiss et al. [102] reported a similar definition of *K*_d_, where [M] was used instead of Σ[M’]. However, as the proton content is experimentally not controlled when *K*_d_ values are measured, not only for the ligand but also for the metal ion, we think that Equation (7) allows a more rigorous calculation of *K*_d_ values. Literature appears to have preference for measuring and reporting *K*_d_ more than *K*_a_ [313,315]. This is probably due to the chemical usefulness and significance of *K*_d_, as it represents the concentration of free metal ion at which the concentrations of free ligand and of the complexes are equivalent [313]. In the following, *K*_d_ values will be considered instead of *K*_a_ ones.

Values of *K*_d_ can be determined with a much simpler experimental design than that used to obtain log*β*, log*K* and pM [313,315]. This is a crucial advantage when complicated ligands are studied, such as proteins, for which the determination of log*β* or log*K* is practically impossible. Still, *K*_d_ can be computed from Equation (7) if log*β* or log*K* values are available, so that the concentrations of all species existing in solution can be calculated. Therefore, *K*_d_ represents a simple tool and practically the only way to compare metal–ligand and metal–protein complex stabilities each other.

The main disadvantage of *K*_d_ is that it depends not only on pH, *c*_M_ and *c*_L_, but also on the copresence of other ligands or other metal ions [316], as these affect Σ[M′] and Σ[L′] in Equation (7) (the same applies for *K*_a_ in equation 6, too). This explains at least in part why reported *K*_d_ values are scarcely reproducible (see Table 1) and depend on the experimental conditions [12]. Equations to correct *K*_d_ values, by taking into account the effect of a competing ligand (e.g., the buffer) and of the different pH, have been proposed [316]. Standardized conditions to measure the *K*_d_ of metal–protein complexes are also being proposed [317,318,319], and this should lead to more reproducible results, thus eventually allowing a more reliable comparison among *K*_d_ numbers.

## 5. The Metal–Ligand Speciation of Anti-Parkinson Drugs

Appendix A reports the metal–ligand speciation available in the literature for the ligands listed in Table 2 (rows) and the relevant metal ions, i.e., Cu(II), Cu(I), Fe(III), Fe(II), Mn(II), and Zn(II) (columns). If not differently specified in the notes of Appendix A, speciation information (stoichiometries of the complexes, and stability constants given as log*β*) has been obtained from the IUPAC stability constant database [320].

The ionic product of water, the stability constants of the metal ion hydrolysis products, and the acidity constants of each ligand, have to be considered to complete the speciation picture and allow speciation calculations. The ionic product of water and the stability constants for hydrolysis products of the considered metal ions are resumed in Appendix A (it is worth noting that these values are only sometimes reported in papers dealing with metal speciation). The acidity constants of the ligands listed in Table 2 have also been taken from the IUPAC stability constant database [320] or from the papers reported in the notes, and they are given as log*β* values in Appendix A (column marked ‘H^+^’).

For some metal–ligand complexes, and for many ligand acidity constants, more than one speciation set has been reported in literature, and/or different log*β* values were proposed. For example, 27 different speciation models have been obtained for the Cu(II)/L-Dopa complexes [320] In Appendix A only one speciation set has been reported, obtained at ionic strength and temperature as close as possible to 0.1 mol/L and 25 °C, respectively. This ionic strength represents a reasonable physiological environment; as regards temperature, 37 °C would better resemble physiological conditions, but speciation data at this temperature are few. For comparison purposes, we preferentially reported data at the most studied temperature of 25 °C. Notes were added in Appendix A if the studied temperature and ionic strength were different than 25 °C and 0.1 mol/L, respectively.

For many other ligands listed in Table 2, no metal speciation set, and sometimes also no acidity constants, were available. This can be ascribed to several reasons, which for some CAs include their very recent development or proposal. For example, 3-(7-Amino-5-(cyclohexylamino)-[1,2,4]triazolo[1,5-*a*][1,3,5]triazin-2-yl)-2-cyanoacrylamide, Aromadendrin, Astilbin, and many other CAs listed in Table 2 have been proposed for MCT in PD only approximately (or less than) one year ago. Possibly, the absence of equilibrium constants can also be justified by the above-mentioned experimental difficulties associated to accurate speciation measurements, which for complicated and often poorly water-soluble molecules may become formidable. Nevertheless, it is possible to tentatively predict the metal speciation of such ligands, by individuating the chelating moiety which is responsible for the complex formation (see also Figure 1), and by considering a simpler ligand having a known metal speciation and bearing the same moiety: ligands having the same chelating functional groups are expected to have a similar metal speciation. For example, complicated molecules bearing a 1,2-diaminoethane chelating group have been considered to have the same metal speciation as 1,2-diaminoethane itself. This predicted speciation should be employed with caution, because inductive and steric effects (and especially resonance ones, if existing) of the remaining part of the molecule might significantly modify the speciation picture. However, these estimations should represent the most reliable values available, until dedicated experimental measurements will be performed. Whenever this kind of assignation has been done, the reported metal-speciation has been marked as “tentative” in the notes of Appendix A. A speciation prediction has also been attempted for molecules bearing two or a maximum of three chelating moieties, by considering their speciation to be similar to that of a simpler ligand bearing the moiety forming the most stable complexes. In these cases, however, the inaccuracy of the predicted speciation might be relatively large.

Table 3 reports selected speciation information regarding Cu(II) which has been computed from the data of Appendix A. Only the ligands for which a Cu(II) speciation was available or has been tentatively estimated were reported in Table 3. Calculations have been performed by the software PITMAT (see [70] and references therein). The first reported value is that of pM (pCu(II) = –log[Cu(II)], which has been computed at pH = 7.4, at *c*_M_ = 10^−6^ mol/L and at *c*_L_ = 10^−5^ mol/L, as recommended in MCT modeling [70,314]. Besides pM, *K*_d_ also was computed according to Equation (7). To the best of our knowledge, no reference pH, *c*_M_ and *c*_L_ values have been hitherto adopted for the calculation of *K*_d_. We propose here that this calculation should be performed at the same conditions as for pM: pH = 7.4, *c*_M_ = 10^−6^ mol/L, and *c*_L_ = 10^−5^ mol/L. The last column of Table 3 reports the most abundant metal complex existing for each ligand at these same physiologically relevant conditions; if available, the charge of this complex is reported as well. Table 4, Table 5, Table 6, Table 7, Table 8 report the same information computed for Cu(I) (Table 4), Fe(III) (Table 5), Fe(II) (Table 6), Mn(II) (Table 7), and Zn(II) (Table 8).

## 6. Possible Usages of Speciation Data for Metal Chelation Therapy against Parkinson’s Disease

Speciation calculations allow to predict which metal and ligand species exist in solution at a given pH and metal and ligand total concentrations. A speciation model can therefore be obtained: this was done for the calculations performed in Table 3, Table 4, Table 5, Table 6, Table 7 and Table 8, where only some information has been given.

As anticipated, values of pM and *K*_d_ are useful to compare the relative strength of the complexes formed by different ligands with the same metal ion, or, conversely, by different metal ions with the same ligand. For example, the complexes formed by Fe(III) with 3-hydroxy-4(1H)-pyridinone (Deferiprone) have larger pM and lower *K*_d_ values than those formed with Luteolin, thus Fe(III)-Deferiprone complexes are stronger than Fe(III)-Luteolin ones. The values of pFe(III) for Deferiprone (19.3) or for Desferrioxamine (26.8) are often considered as milestones when new CAs are proposed for the chelation therapy of Fe overload [314]: new compounds are considered to be effective enough if their pFe(III) is larger than that of Deferiprone or of Desferrioxamine.

More importantly, *K*_d_ values can be compared with those reported in Table 1, allowing the assessment of whether a given ligand is able to remove a metal ion from α-syn: removal can occur if the *K*_d_ value is lower than that of α-syn. This approach has been proposed for MCT in Alzheimer’s disease, where *K*_d_ values of the CA + metal ion complexes were recommended to be 10–100 times lower than *K*_d_ values of the amyloid β protein + metal ion ones [316]. If the same approach is adopted for PD, it can be, e.g., deduced that Desferrioxamine (*K*_d_ = 1.81 × 10^−7^) is able to remove Fe(III) from α-syn (*K*_d_ = 10^−4^), and L-Dopa (*K*_d_ = 7.95 × 10^−6^) is able to remove Cu(II) from α-syn (*K*_d_ = 10^2^). Many other ligands, including L-Dopa (*K*_d_ = 2.96 × 10^4^), cannot remove Fe(III) from α-syn. However, it is necessary to underline that the *K*_d_ values reported in Table 1 are not completely reliable (the same also applies for *K*_d_ values of amyloid β protein + metal ion [12]), for the reasons stated above, so this approach does not (still) allow drawing definite conclusions about the removal of the relevant metal ion from proteins. On the other hand, the control of metal ion dyshomeostasis in PD requires that the CAs do not form too strong metal complexes to avoid metal anemia and allow metal redeployment to other compartments, according to the conservative chelation strategy. Too high or too low *K*_d_ values are thus not suitable. Unfortunately, the limiting *K*_d_ values which an ideal CA should possess to be employed for the PD therapy are not known.

The information regarding the most abundant complex existing at physiological conditions can be useful for two reasons. The identity of the existing complexes (and in particular of the most abundant one) and their charge are crucial in determining their redistribution once the target metal ion has been complexed by the CA. For example, a charged complex is expected to be hydrophilic, thus being unable to pass cellular barriers and preferring to be solubilised in aqueous solutions (e.g., in the blood), whereas neutral species should behave in an opposite way. The structure of the complexes, which might be deduced from the stoichiometry, also has a main role in determining their properties and toxicity [321].

As regards Fe and Cu, it is necessary to consider that they can undergo redox reactions even when complexed by a ligand. These reactions might be as harmful as or even more dangerous than those caused by the target metal ion at pathological in vivo conditions. The redox-induced toxicity of Fe and Cu complexes formed by several ligands is well known, so it has been used to develop new anti-cancer drugs [322], but it appears to have been generally overlooked when MCT is employed for PD. The redox activity of Fe and Cu complexes depends on the relative stability of the complexes formed by the ions at the two oxidation states. As regards Fe, if a ligand L forms the complexes Fe^III^L and Fe^II^L with Fe(III) and Fe(II), respectively, a redox half-reaction can occur:Fe^III^L + e^−^ = Fe^II^L(8)

By means of simple substitutions in the Nernst equation of Fe(III)/Fe(II), the standard reduction potential of (8) can be computed:(9)$$E_{Fe^{III}L/Fe^{II}L}^{0}=E^{0}{}_{Fe\left(III\right)/Fe\left(II\right)}+\frac{RT}{F}\ln\frac{\beta_{Fe^{II}L}}{\beta_{Fe^{III}L}}$$
where *E*^0^_Fe(III)/Fe(II)_ is the standard reduction potential for free Fe (0.771 V), and *β*_Fe^III^__L_ and *β*_Fe^II^__L_ are the cumulative stability constants of Fe^III^L and Fe^II^L, respectively. Values of *E*^0^ of any Fe or Cu complexes can be derived in a similar way if metal–ligand speciation at both oxidation states is known. Alternatively, electrochemical values can experimentally be obtained from voltammetric measurements (see, e.g., [323]). The standard reduction potentials of Fe complexes might have an important role in determining whether they can undergo harmful redox cycling in vivo, as extensively descrived by Merkofer et al. [324] and recently reviewed by Koppenol and Hider [325]: in general, it appears that negative *E*^0^ values can guarantee the absence of such toxic phenomena. However, further work is necessary to evaluate limiting Fe and Cu *E*^0^ values under which no redox damage occurs in PD brains.

Other possible information which can be gathered from speciation calculations, even if focused on the bloodstream, has recently been reviewed by Kiss et al. [326,327].

## 7. Concluding Remarks

The development of drugs able to target several pathological pathways appears to be the best approach for PD therapy and of other important NDs such as Alzheimer’s disease and Amyotrophic Lateral Sclerosis. Compounds which form complexes with the PD relevant metal ions, i.e., Cu(II), Cu(I), Fe(III), Fe(II), Mn(II) and Zn(II), are aimed to target metal dyshomeostasis. For these CAs and for these metal ions, the knowledge of metal–ligand speciation is of primary importance to predict the efficacy of the CA, its ability to remove dysregulated metal ions from toxic storages such as the α-syn complex and redeploy metal ions to safe stores (conservative chelation), the possible toxic effects induced by the metal complexes formed in PD brain, and in general to be able to model the distribution of the metal–ligand species in vivo. Still much work has to be performed to define the upper and the lower limiting metal *K*_d_ values required by a CA to disrupt the α-syn complex without causing excessive metal removal, as well as the suitable standard reduction potentials required by the complexes to avoid harmful redox cycling in the brain. Also, available speciation information is in part lacking, especially as regards Cu(I), for which very few stability constants values have been hitherto determined (possible strategies for effective studies of Cu(I) speciations have been proposed [328]), but also for other metal ions and for several complicated or recently proposed CAs. If a complete metal–ligand speciation study (aimed to determine stability constants) cannot be performed, a *K*_d_ value should at least be determined. This amount represents a key number which can be used to compare simple metal–ligand complexes, for which the full speciation picture is available, with complicated ones like those involving α-syn, for which this information cannot be obtained. However, standardized experimental procedures are recommended to allow *K*_d_ values to be more rigorously and reliably compared with each other.

## Figures and Tables

**Figure 1 biomolecules-09-00269-f001:**
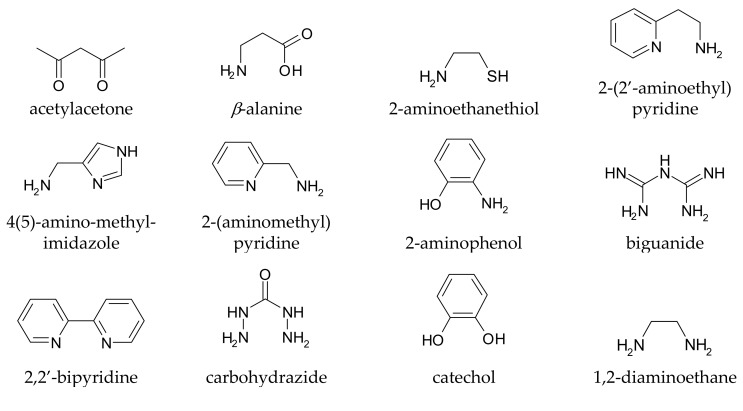
Simple metal-chelating moieties (listed in alphabetical order) which are often encountered in molecules used or proposed for the metal chelation therapy in Parkinson’s disease.

**Table 1 biomolecules-09-00269-t001:** Dissociation constants (*K*_d_) obtained at physiological pH for the complexes formed between metal ions and α-synuclein (values computed for the highest affinity binding site), and references (reviews).

Metal Ion	*K*_d_ (nmol/L)	References
Cu(II)	10^2^	[12,61]
Cu(I)	10^4^–10^3^	[61]
Fe(III)	10^−4^	[12,60]
Fe(II)	10^6^–5 × 10^4^	[12,60]
Mn(II)	10^6^	[60]
Zn(II)	>10^6^	[61]

**Table 2 biomolecules-09-00269-t002:** Compounds displaying metal-chelation properties which have been used, tested or proposed for the therapy against PD, as obtained from a literature survey in the year range 2014–2019 (April). Substances are listed in the first column according to their alphabetical order. Only the latest and/or the most important references (e.g., reviews) are given in the last column. This Table is a subset of Appendix A, which also includes non-chelating compounds and compounds with non-predictable metal–ligand speciation. The chemical structure of each substance is reported in Appendix A.

Compound Name(s)	References
7DH	[112]
7MH	[112]
8A	[89]
8B	[89]
8C	[89]
8E	[89]
8F	[89]
*N*-Acetylcysteine	[113,114]
ACPT-I	[115]
ADX88178	[116]
Alaternin	[117]
Alvespimycin	[118]
AM-251	[119,120]
Ambroxol	[121,122]
3-(7-Amino-5-(cyclohexyl-amino) -[1,2,4]triazolo[1,5-a] [1,3,5]triazin-2-yl)-2 -cyanoacrylamide	[123]
Aminothiazoles derivatives as SUMOylation activators	[124]
AMN082	[115]
Amodiaquine	[125,126]
Antagonist of the A(2A) adenosine receptor-derivative 49	[127]
Apigenin	[128,129,130,131]
Apomorphine	[132,133]
l-Arginine	[134]
Aromadendrin	[128]
Ascorbic acid	[135,136]
ASI-1	[12]
ASI-5	[12]
Astilbin	[137]
Azilsartan	[138]
Baicalein	[139,140,141]
Benserazide	[142,143]
7*H*-Benzo, perimidin-7-one derivatives (R6 = OH)	[144]
4*H*-1-Benzopyran-4-one	[145]
8-Benzyl-tetrahydropyrazino, purinedione derivatives (derivative n.57)	[146]
Bikaverin	[147]
(−)-*N*6-(2-(4-(Biphenyl-4-yl) piperazin-1-yl)-ethyl)-*N*6- propyl-4,5,6,7-tetrahydro- benzo, thiazole-2,6-diamine derivatives	[148]
2.2’-Bipyridyl (2,2’-bipyridine)	[112]
4-((5-Bromo-3-chloro-2- hydroxybenzyl) amino)-2- hydroxybenzoic acid (LX007, ZL006)	[149,150]
C-3 (α carboxyfullerene)	[151]
Caffeic acid amide analogues	[152,153,154,155]
Carbazole-derived compounds	[156]
Carbidopa	[135,157]
Carnosic acid	[154,158]
Catechin	[24,128]
Ceftriaxone	[12,159,160,161]
Celastrol	[162,163]
CEP-1347	[164,165]
Chebulagic acid	[166]
Chlorogenic acid	[167]
3′-*O*-(3-Chloropivaloyl) quercetin	[168]
Chlorpromazine	[108]
Chrysin	[128,169,170]
Clioquinol	[89,91,171,172]
Clioquinol-selegiline hybrid	[79]
Clovamide analogues (R1 and R2 = OH, and/or R3 and R4 = OH)	[173]
“Compound 1”	[174]
“Compound (−)-8a”	[175]
“Compound 8”	[176]
“Compound 21”, derivative of 3-methyl-1-(2,4,6-trihydroxy phenyl) butan-1-one	[177]
“Compound (−)-21a” , derivative of *N*-6-(2-(4-(1*H*-indol-5-yl) piperazin-1-yl)ethyl)-*N*-6- propyl-4,5,6,7-tetrahydro- benzo[*d*]thiazole-2,6-diamine	[178]
Creatine	[179,180]
Cudraflavone B	[181]
Curcumin	[89,117,182,183,184]
Cyanidin	[185,186]
D-512	[187]
D-607 (bipyridyl-D2R/D3R agonist hybrid)	[12,188,189]
DA-2 (8D)	[12,89]
DA-3	[12]
DA-4	[12]
Dabigatran etexilate	[190]
Dabrafenib	[191]
(*S*)-3,4-DCPG	[115]
Deferasirox	[24]
Deferricoprogen	[192]
Delphinidin	[160,185,193,194]
Demethoxycurcumin	[195]
Dendropanax morbifera active compound	[196]
Desferrioxamine (Desferoxamine, Desferal, DFO)	[112]
(S)-*N*-(3-(3,6-Dibromo-9*H* -carbazol-9-yl)-2-fluoropropyl)-6-methoxypyridin-2-amine	[197]
4,5-*O*-Dicaffeoyl-1-*O*-(malic acid methyl ester)-quinic acid derivatives (R1, R2, R3, R4, or R5 = caffeoyl)	[198]
Dihydromyricetin	[199]
5-(3,4-Dihydroxybenzylidene) -2,2-dimethyl-1,3-dioxane-4,6-dione	[200]
7,8-Dihydroxycoumarin derivative DHC12	[79]
3′,4′-Dihydroxyflavone	[201]
7,8-Dihydroxyflavone	[202,203]
5,7-Dihydroxy-4′-methoxy flavone	[204]
(*E*)-3,4-Dihydroxystyryl aralkyl sulfones	[205]
(*E*)-3,4-Dihydroxystyryl aralkyl sulfoxides	[205]
5,3’-Dihydroxy-3,7,4’ -trimethoxyflavone	[206]
2-[[(1,1-Dimethylethyl) oxidoimino]-methyl]-3,5,6- trimethylpyrazine	[207]
DKP	[85]
L-DOPA (levodopa, CVT-301)	[132,135,208]
DOPA-derived peptido-mimetics (deprotected)	[209]
DOPA-derived peptido-mimetics (protected)	[209]
L-DOPA deuterated (D3-L-DOPA)	[210]
Doxycycline	[211,212]
Droxidopa	[110]
Echinacoside	[213]
Ellagic acid	[214]
Entacapone (Comtan, ASI-6)	[12,215,216]
Enzastaurin	[164]
Epicatechin	[128,160,193,194]
Epigallocatechin-3-gallate	[117,217,218]
Etidronate (HEDPA)	[219]
Exifone	[12]
F13714	[220]
F15599	[220,221]
Farrerol	[222]
Fisetin (3,3′,4′,7-Tetrahydroxy flavone)	[223,224,225]
Fraxetin	[117]
Galangin	[226]
Gallic acid and derivatives	[214,227,228]
Gallocatechin	[128]
Garcinol	[229]
Genistein	[117,128,230,231]
Glutamine	[232]
Glutathione derivatives	[63]
Glutathione -hydroxyquinoline compound	[233]
Glutathione-l-DOPA compound	[234]
Gly-N-C-DOPA	[209]
GSK2795039	[108]
Guanabenz	[235]
Hesperidin	[128,236]
Hinokitiol	[237]
8-HQ-MC-5 (VK-28)	[12,24,89,92,93]
4-Hydroxyisophthalic acid	[238]
1-Hydroxy-2-pyridinone derivatives	[89,239]
3-Hydroxy-4(1*H*)-pyridinone (Deferiprone)	[112,239,240]
8-Hydroxyquinoline	[241]
8-Hydroxyquinoline-2- carboxaldehyde isonicotinoyl hydrazone	[242]
Hydroxyquinoline-propargyl hybrid (HLA 20)	[12,79,89]
Hydroxytyrosol butyrate	[243]
Hyperoside	[117]
IC87201	[150]
Icariin	[244]
Icariside II	[245]
l-(7-imino-3-propyl-2,3- dihydrothiazolo [4,5-*d*]pyrimidin-6(7*H*)-yl)urea	[246]
Imipramine	[247]
Isobavachalcone	[248]
Isochlorogenic acid	[167]
Isoquercetin (Isoquercitrin)	[249]
Kaempferol	[128,160,193,194]
Kaempferol, 3-*O*-a-l arabinofuranoside-7-*O*-a-l-rhamnopyranoside	[214]
KR33493	[250]
Kukoamine	[251]
Lestaurtinib	[164]
Lipoic acid	[252,253,254]
Luteolin	[128]
LY354740	[115]
M10	[24]
M30 (VAR10303)	[112,255]
M99	[24]
Macranthoin G	[256]
Magnesium lithospermate B	[257]
α-Mangostin	[258]
γ-Mangostin	[259]
MAOI-1	[12]
MAOI-2	[12]
MAOI-4	[12]
MAOI-8	[12]
Meclofenamic acid	[260]
Metformin (Met)	[261,262]
Methoxy-6-acetyl-7- methylijuglone	[117]
N’-(4-Methylbenzylidene)-5-phenylisoxazole-3-carbohydrazide	[263]
Mildronate	[264]
Minocycline	[12,160,265]
Mitomycin C	[266]
MitoQ	[108]
Morin (3,5,7,29,49-pentahydroxy flavone)	[160,193,194,267]
[18F]MPPF	[107]
MSX-3	[268]
Myricetin	[128,269,270]
Myricitrin	[271]
Naringenin	[128,272]
Naringin	[117,273,274]
Nicotinamide adenine dinucleotide phosphate (NADPH)	[275,276]
Nicotinamide mononucleotide	[277]
Nitecapone	[278]
Nordihydroguaiaretic	[160,193,194]
Oleuropein	[279]
Opicapone	[278]
P7C3	[280,281]
PBF-509	[282]
PBT2	[89,283]
PBT434	[284]
Petunidin	[185]
Phenothiazine 2Bc (n = 0 and n = 1)	[285]
Phenylhydroxamates	[286]
Piceatannol	[160,193,194]
Pinostrobin	[287]
Piperazine-8-OH-quinolone hybrid	[79]
Preladenant	[282,288]
Promethazine	[289]
Protocatechuic acid	[170]
Protosappanin A	[290]
Punicalangin	[270,291]
Pyrazolobenzothiazine-based carbothioamides	[292]
Pyridoxal isonicotinoyl hydrazone (PIH) and related compounds	[24,89]
Pyrimidinone 8	[293]
Q1	[89]
Q4	[89]
Quercetin	[117,294,295]
Quinolines Derivatives as SUMOylation activators	[124]
Radotinib	[296]
Riboflavin	[297,298]
Rifampicin (ASI-3)	[12,160]
Rimonabant	[119,120,282]
Rosmarinic acid	[154,299]
Rotigotine	[105,133,278,300]
Rutin	[128,249]
Salicylate, sodium salt	[301]
Salvianolic Acid B	[117]
SCH-58261	[105,302]
SCH412348	[105]
Silibinin (silybin) A, B	[89]
Silydianin	[24]
ST1535	[282]
ST4206	[282]
Staurosporine	[164]
Stemazole	[303]
Sulfuretin	[304]
Tannic acid	[160,193,194]
Tanshinol	[117]
Taurine	[305]
Taxifolin	[128]
Tectorigenin	[306]
Tetracycline	[307]
Tolcapone (ASI-7)	[12,105]
Tozadenant	[282,302]
Transilitin	[270]
O-Trensox	[24]
2′,3′,4′-Trihydroxyflavone	[270]
2,3,3-Trisphosphonate	[219]
V81444	[282]
VAS3947	[108]
VAS2870	[108]
Verbascoside	[160,193,194]
WIN 55,212-2	[119,120]
WR-1065	[308]
Zonisamide	[309,310,311]

**Table 3 biomolecules-09-00269-t003:** pCu(II) and *K*_d_ values, and the most abundant Cu(II) complex, obtained at physiologically relevant conditions: pH = 7.4, *c*_Cu_ = 10^–6^ mol/L, and *c*_L_ = 10^–5^ mol/L. Computations have been performed from data of Appendix A. Refer to Appendix A to identify ligands for which only tentative speciations were proposed, and ligands whose complexes have unknown charges.

Compound Name(s)	pCu(II)	*K*_d_ (nmol/L)	Most Abundant Complex
7DH 7MH	14.2	5.91 × 10^−5^	CuL_2_
8A 8B 8C	14.2	5.91 × 10^−5^	CuL_2_
8E 8F	10.6	2.35 × 10^−1^	CuL_2_
*N*-Acetyl cysteine	6.2	3.82 × 10^4^	CuL
ACPT-I	7.3	5.93 × 10^2^	CuL
ADX88178	7.4	5.15 × 10^2^	CuL
Alaternin	16.5	3.39 × 10^−7^	CuL_2_
Alvespimycin	10.3	5.35 × 10^−1^	CuL_2_^2+^
AM-251	6.3	1.47 × 10^4^	CuL
Ambroxol	9.2	7.26	CuL^+^
3-(7-Amino-5-(cyclohexylamino)-[1,2,4]triazolo[1,5-*a*][1,3,5] triazin-2-yl)-2-cyanoacrylamide	9.1	7.58	CuL_2_
Aminothiazoles derivatives as SUMOylation activators	9.7	1.86	CuL_2_
AMN082	10.3	5.35 × 10^−1^	CuL_2_^2+^
Amodiaquine	6.1	1.90 × 10^7^	CuHL
Antagonist of the A(2A) adenosine receptor-derivative 49	6.4	9.65 × 10^3^	CuL
Apigenin	6.8	2.39 × 10^3^	CuH_2_L^+^
Apomorphine	7.4	5.06 × 10^2^	CuL
l-Arginine	7.1	9.57 × 10^2^	CuL^2+^
Aromadendrin	11.1	8.29 × 10^−2^	CuL
Ascorbic acid	6.1	2.43 × 10^5^	Cu_2_H_–2_L_2_
ASI-1	7.9	1.21 × 10^2^	CuL_2_
ASI-5	6.1	2.23 × 10^6^	CuL
Astilbin	7.4	5.06 × 10^2^	CuL
Azilsartan	6.1	4.53 × 10^5^	CuL
Baicalein	9.3	5.41	CuL_2_^2−^
Benserazide	9.3	5.30	CuL_2_
7*H*-benzo[e] perimidin-7-one derivatives	14	1.05 × 10^−4^	CuL_2_
8-Benzyl-tetrahydropyrazino[2,1-*f*]purinedione (derivative n. 57)	6.1	3.36 × 10^6^	CuL
Bikaverin	14	1.05 × 10^−4^	CuL_2_
(−)-*N*6-(2-(4-(Biphenyl-4-yl)piperazin-1-yl)-ethyl)-*N*6-propyl-4,5,6,7-tetrahydrobenzo[*d*]thiazole-2,6-diamine derivatives	10.3	5.35 × 10^−1^	CuL_2_^2+^
2,2′-bipyridyl	10.6	2.35 × 10^−1^	CuL_2_^2+^
4-((5-bromo-3-chloro-2-hydroxybenzyl) amino)-2-hydroxybenzoic acid (LX007, ZL006)	6.2	6.51 × 10^4^	CuL
C-3 (α-carboxyfullerene)	6.9	1.96 × 10^3^	CuL
Caffeic acid amide analogues	7.3	6.02 × 10^2^	CuH_−1_L
Carbazole-derived compounds	10.3	5.35 × 10^−1^	CuL_2_^2+^
Carbidopa	15.1	8.11 × 10^−6^	CuH_−2_L
Carnosic acid	7.4	5.06 × 10^2^	CuL
Catechin	7.9	1.50 × 10^2^	CuH_2_L
Ceftriaxone	6.1	4.04 × 10^6^	CuL
Celastrol	6.5	5.27 × 10^3^	CuL
Chebulagic acid	6.3	1.51 × 10^4^	CuHL
Chlorogenic acid	8.3	5.04 × 10^1^	CuL^−^
3′-*O*-(3-chloropivaloyl) quercetin	11.1	8.29 × 10^−2^	CuL
Chlorpromazine	6.3	1.47 × 10^4^	CuL^2+^
Chrysin	10.6	2.97 × 10^−1^	CuHL^+^
Clioquinol	14.2	5.91 × 10^−5^	CuL_2_
Clovamide analogues (R_1_ and R_2_ = OH, and/or R_3_ and R_4_ = OH)	7.4	5.06 × 10^2^	CuL
“Compound 1”	10.3	5.35 × 10^−1^	CuL_2_^2+^
“Compound 8”	6.1	4.37 × 10^5^	CuL_2_
“Compound 21”, derivative of 3-methyl-1-(2,4,6-trihydroxyphenyl) butan-1-one	7.3	5.93 × 10^2^	CuL^+^
“Compound (−)-21a”, derivative of *N*-6-(2-(4-(1*H*-indol-5-yl) piperazin-1-yl)ethyl)-*N*-6-propyl-4,5,6,7-tetrahydrobenzo[*d*] thiazole-2,6-diamine	10.3	5.35 × 10^−1^	CuL_2_^3+^
Creatine	6.8	2.26 × 10^3^	CuH_−1_L
Cudraflavone B	11.1	8.29 × 10^−2^	CuL
Curcumin	7.9	1.21 × 10^2^	CuL_2_
Cyanidin	7.4	5.06 × 10^2^	CuL
D512	10.3	5.35 × 10^−1^	CuL_2_^2+^
D607 (bipyridyl-D2R/D3R agonist hybrid)	10.6	2.35 × 10^−1^	CuL_2_
DA-2 (8D)	14.2	5.91 × 10^−5^	CuL_2_
DA-3	10.3	5.35 × 10^−1^	CuL_2_
DA-4	10.3	5.35 × 10^−1^	CuL_2_
Dabigatran etexilate	10.3	5.35 × 10^−1^	CuL_2_
Dabrafenib	6.1	3.36 × 10^6^	CuL
(*S*)-3-4-DCPG	6.1	4.54 × 10^5^	CuL
Deferricoprogen	12.6	3.17 × 10^−3^	CuHL
Delphinidin	9.3	5.30	CuL_2_
Demethoxycurcumin	7.9	1.21 × 10^2^	CuL_2_
Dendropanax morbifera active compound	7.4	5.06 × 10^2^	CuL
Desferrioxamine (Deferoxamine, Desferal, DFO)	11.4	4.98 × 10^−2^	CuH_2_L^+^
(S)-*N*-(3-(3-6-dibromo-9H-carbazol-9-yl)-2-fluoropropyl)-6- methoxypyridin-2-amine	6.3	1.47 × 10^4^	CuL
4, 5-*O*-Dicaffeoyl-1-*O*-(malic acid methyl ester)-quinic acid (R_1_, R_2_, R_3_, R_4_, or R_5_ = caffeoyl)	7.4	5.06 × 10^2^	CuL
Dihydromyricetin	9.3	5.30	CuL_2_
5-(3,4-Dihydroxybenzylidene)-2,2-dimethyl-1,3-dioxane-4,6-dione	7.4	5.06 × 10^2^	CuL
7,8-Dihydroxycoumarin derivative DHC12	7.4	5.06 × 10^2^	CuL
3′,4′-Dihydroxyflavone	7.4	5.06 × 10^2^	CuL
7,8-Dihydroxyflavone	7.4	5.06 × 10^2^	CuL
5,7-Dihydroxy-4′-methoxyflavone	6.6	4.58 × 10^3^	CuL
(*E*)-3, 4-Dihydroxystyryl aralkyl sulfones	7.4	5.06 × 10^2^	CuL
(*E*)-3, 4-Dihydroxystyryl aralkyl sulfoxides	7.4	5.06 × 10^2^	CuL
5,3′-Dihydroxy-3,7,4′-trimethoxyflavone	6.6	4.58 × 10^3^	CuL
2-[[(1,1-Dimethylethyl) oxidoimino]-methyl]-3,5,6-trimethylpyrazine	9.6	2.98	CuL
DKP	8.1	8.43 × 10^1^	CuL
l-DOPA (levodopa, CVT-301)	15.2	7.95 × 10^−6^	CuH_−2_L^3−^
DOPA-derived peptido-mimetics (deprotected)	15.2	7.95 × 10^−6^	CuH_−2_L_2_
DOPA-derived peptido-mimetics (protected)	7.4	5.06 × 10^2^	CuL
l-DOPA deuterated	15.2	7.95 × 10^−6^	CuH_−2_L^3−^
Doxycycline	8.9	1.28 × 10^1^	CuL
Droxidopa	15.2	7.95 × 10^−6^	CuH_−2_L^3−^
Echinacoside	7.4	5.06 × 10^2^	CuL
Ellagic acid	7.4	5.06 × 10^2^	CuL
Entacapone (comtan, ASI-6)	10.1	8.00 × 10^−1^	CuL_2_^2−^
Enzastaurin	10.3	5.35 × 10^−1^	CuL_2_
Epigallocatechin-3-gallate	6.1	1.33 × 10^5^	CuH_2_L_2_
Etidronate (HEDPA)	9	1.11 × 10^1^	CuL^2−^
Exifone	6.3	1.51 × 10^4^	CuHL
F13714, F15599	9.7	1.86	CuL_2_
Farrerol	11.1	8.29 × 10^−2^	CuL
Fisetin (3,3′,4′,7-tetra-hydroxy-flavone)	7.4	5.06 × 10^2^	CuL
Fraxetin	7.4	5.06 × 10^2^	CuL
Galangin	9.1	8.20	CuL
Gallic acid derivatives	6.3	1.51 × 10^4^	CuHL^−^
Gallocatechin	9.3	5.30	CuL_2_
Garcinol	7.4	5.06 × 10^2^	CuL
Genistein	11.1	8.29 × 10^−2^	CuL
Glutamine	7.3	5.38 × 10^2^	CuL^+^
Glutathione derivatives	6.2	7.23 × 10^4^	CuL^−^
Glutathione-hydroxy-quinoline compound	9.4	5.00	CuH_−1_L^+^
Glutathione-l-DOPA compound	13.5	3.98 × 10^−4^	CuH_−1_L
Gly-N-C-DOPA	15.2	7.95 × 10^−6^	CuH_−2_L^3−^
GSK2795039	12	1.02 × 10^−2^	CuL_2_
Guanabenz	10.3	5.35 × 10^−1^	CuL_2_
Hesperidin	11.1	8.29 × 10^−2^	CuL
Hinokitiol	7.3	6.69 × 10^2^	CuL^+^
8-HQ-MC-5 (VK-28)	14.2	5.91 × 10^−5^	CuL_2_
4-Hydroxyisophthalic acid	6.3	2.06 × 10^4^	CuL
1-Hydroxy-2-pyridinone derivatives	8.4	4.26 × 10^1^	CuL_2_
3-Hydroxy-4(1*H*)pyridinone (Deferiprone)	10.2	6.60 × 10^−1^	CuL_2_
3-Hydroxy-4(1*H*)pyridinone derivatives (R = H)	10.2	6.60 × 10^−1^	CuL_2_
8-Hydroxyquinoline	14.2	5.91 × 10^−5^	CuL_2_
8-Hydroxyquinoline-2-carboxaldehyde isonicotinoyl hydrazone	14.2	5.91 × 10^−5^	CuL_2_
Hydroxy-quinoline-propargyl hybrids (HLA20)	14.2	5.91 × 10^−5^	CuL_2_
Hydroxytyrosol butyrate	7.4	5.06 × 10^2^	CuL
Hyperoside	9.1	8.20	CuL
IC87201	6.1	1.90 × 10^7^	CuHL
Icariin	6.6	4.58 × 10^3^	CuL
Icariside II	6.6	4.58 × 10^3^	CuL
l-(7-Imino-3-propyl-2,3-dihydrothiazolo[4,5-*d*]pyrimidin -6(7*H*)-yl)urea	6.1	4.37 × 10^5^	CuL_2_
Imipramine	6.3	1.47 × 10^4^	CuL^2+^
Isobavachalcone	6.2	8.13 × 10^4^	CuL^+^
Isochlorogenic acid	8.3	5.04 × 10^1^	CuL^−^
Isoquercetin (isoquercitrin)	9.1	8.20	CuL
Kaempferol	9.1	8.20	CuL
KR33493	7.3	5.93 × 10^2^	CuL
Kukoamine	7.4	5.06 × 10^2^	CuL
Lestaurtinib	10.3	5.35 × 10^−1^	CuL_2_
Lipoic acid	6.1	4.41 × 10^5^	CuL^+^
Luteolin	7.4	5.06 × 10^2^	CuL
M10 M30 (VAR10303) M99	14.2	5.91 × 10^−5^	CuL_2_
Macranthoin G	7.4	5.06 × 10^2^	CuL
Magnesium lithospermate B	7.4	5.06 × 10^2^	CuL
*α*-mangostin	7.4	5.06 × 10^2^	CuL
*γ*-Mangostin	7.4	5.06 × 10^2^	CuL
MAOI-1	9.3	5.30	CuL_2_
MAOI-2	10.3	5.35 × 10^−1^	CuL_2_
MAOI-4	6.3	1.47 × 10^4^	CuL
MAOI-8	6.1	1.90 × 10^7^	CuHL
Metformin (met)	6.3	1.61 × 10^4^	CuL^+^
Methoxy-6-acetyl-7-methylijuglone	14	1.05 × 10^−4^	CuL_2_
*N*′-(4-methylbenzylidene)-5-phenylisoxazole-3-carbohydrazide	6.3	1.38 × 10^4^	CuL
Minocycline	11.6	2.83 × 10^−2^	CuL
Mitomycin C	7.1	9.14 × 10^2^	CuL
Morin	6.1	2.27 × 10^15^	CuH_3_L
[18F]MPPF	10.3	5.35 × 10^−1^	CuL_2_
MSX-3	6.1	2.01 × 10^6^	CuL
Myricetin Myricitrin	9.1	8.20	CuL
Naringin	6.9	1.58 × 10^3^	CuHL
Naringenin	6.9	1.58 × 10^3^	CuHL
Nicotinamide adenine dinucleotide phosphate (NADPH)	6.4	8.91 × 10^3^	CuL
Nicotinamide mononucleotide	6.1	2.01 × 10^6^	CuL
Nitecapone	10.1	8.00 × 10^−1^	CuL_2_^2−^
Nordihydroguaiaretic acid	7.4	5.06 × 10^2^	CuL
Oleuropein	7.4	5.06 × 10^2^	CuL
Opicapone	10.1	8.00 × 10^−1^	CuL_2_
P7C3	7.8	3.76 × 10^2^	Cu_2_H_−2_L_2_^+^
PBT2	12.4	4.04 × 10^−3^	CuL^+^
PBT434	11.2	7.21 × 10^−2^	CuL^+^
Petunidin	7.4	5.06 × 10^2^	CuL
Phenothiazine 2Bc (n=0)	10.3	5.35 × 10^−1^	CuL_2_^2+^
Phenothiazine 2Bc (n=1)	6.3	1.47 × 10^4^	CuL^2+^
Phenylhydroxamates	7	1.46 × 10^3^	CuL
Piceatannol	7.4	5.06 × 10^2^	CuL
Pinostrobin (5-hydroxy-7-methoxy-flavone)	6.6	4.58 × 10^3^	CuL^+^
Piperazine-8-OH-quinolone hybrid	14.2	5.91 × 10^−5^	CuL_2_
Preladenant	10.3	5.35 × 10^−1^	CuL_2_
Promethazine	8.2	7.99 × 10^1^	CuL^2+^
Protocatechuic acid	8.1	8.13 × 10^1^	CuL^−^
Protosappanin A	7.4	5.06 × 10^2^	CuL
Punicalangin	8.2	6.89 × 10^1^	CuL
Pyrazolobenzothiazine-based carbothioamides	6.1	4.66 × 10^5^	CuL
Pyrimidinone 8	10.3	5.35 × 10^−1^	CuL_2_
Q1 Q4	14.2	5.91 × 10^−5^	CuL_2_
Quercetin	9.1	8.20	CuL^3−^
Quinoline derivatives SUMOylation activators	7.2	7.26 × 10^2^	CuL^2+^
Radotinib	10.6	2.35 × 10^−1^	CuL_2_
Riboflavin	6.1	1.65 × 10^5^	CuHL^3+^
Rifampicin (ASI-3)	6.5	6.97 × 10^3^	CuL
Rimonabant	8.1	8.43 × 10^1^	CuL
Rosmarinic acid	7.4	5.06 × 10^2^	CuL
Rotigotine	6.1	1.45 × 10^8^	CuL_2_
Rutin	9.1	8.20	CuL
Salicylate, sodium salt	6.3	2.06 × 10^4^	CuL
Salvianolic acid B	7.4	5.06 × 10^2^	CuL
SCH58261 SCH412348	9.1	7.58	CuL_2_
ST1535 ST4206	9.1	7.58	CuL_2_
Staurosporine	10.3	5.35 × 10^−1^	CuL_2_
Stemazole	6.1	4.66 × 10^5^	CuL
Sulfuretin	7.4	5.06 × 10^2^	CuL
Tannic acid	6.1	1.84 × 10^5^	CuL
Tanshinol	7.4	5.06 × 10^2^	CuL
Taurine	6.1	1.23 × 10^7^	CuL^+^
Taxifolin	10.4	4.55 × 10^−1^	CuL^2−^
Tectorigenin	11.1	8.29 × 10^−2^	CuL
Tetracycline	6.4	1.08 × 10^4^	CuL
Tolcapone (ASI-7)	10.1	8.00 × 10^−1^	CuL_2_
Transilitin	7.4	5.06 × 10^2^	CuL
o-Trensox	22.9	1.51 × 10^−13^	CuL^4−^
2′, 3′, 4′-Trihydroxyflavone	9.3	5.30	CuL_2_^2−^
2,3,3-Trisphosphonate	14	9.98 × 10^−5^	CuL_2_
V81444	9.7	1.86	CuL_2_
VAS3947 VAS2870	6.1	4.37 × 10^5^	CuL_2_
Verbascoside	7.3	6.02 × 10^2^	CuH_−1_L
WIN 55, 212-2	10.3	5.35 × 10^−1^	CuL_2_^2+^
WR-1065	6.6	3.67 × 10^3^	CuL^2+^
Zonisamide	7.4	4.62 × 10^2^	CuL

**Table 4 biomolecules-09-00269-t004:** pCu(I) and *K*_d_ values, and the most abundant Cu(I) complex, obtained at physiologically relevant conditions: pH = 7.4, *c*_Cu_ = 10^−6^ mol/L, and *c*_L_ = 10^−5^ mol/L. See caption of Table 3 for other notes.

Compound Name(s)	pCu(I)	*K*_d_ (nmol/L)	Most Abundant Complex
7DH 7MH	6.3	8.38 × 10^3^	CuL_2_
8A 8B 8C	6.3	8.38 × 10^3^	CuL_2_
8E 8F	6.2	1.87 × 10^4^	CuL
ACPT-I	6	2.54 × 10^8^	CuL_2_
ADX88178	7.7	1.77 × 10^2^	CuL
Alvespimycin	6	7.98 × 10^7^	CuL_2_^+^
3-(7-Amino-5-(cyclohexylamino)-[1,2,4]triazolo[1,5-*a*] [1,3,5]triazin-2-yl)-2-cyanoacrylamide	6	7.98 × 10^7^	CuL_2_
Aminothiazoles derivatives as SUMOylation activators	6	2.66 × 10^6^	CuL_2_
AMN082	6	7.98 × 10^7^	CuL_2_
Antagonist of the A(2A) adenosine receptor (derivative 49)	6	3.06 × 10^7^	CuL_2_
8-Benzyl-tetrahydropyrazino[2,1-*f*]purinedione (derivative 57)	7.6	2.17 × 10^2^	CuL
(−)-*N*6-(2-(4-(Biphenyl-4-yl)piperazin-1-yl)-ethyl)-*N*6-propyl-4,5,6,7-tetrahydrobenzo[*d*]thiazole-2,6-diamine derivatives	6	7.98 × 10^7^	CuL_2_^+^
2,2′-bipyridyl	6.2	1.87 × 10^4^	CuL^+^
Carbazole-derived compounds	6	7.98 × 10^7^	CuL_2_^+^
Ceftriaxone	6	2.54 × 10^8^	CuL_2_
Clioquinol	7.2	5.79 × 10^2^	CuL_2_^−^
“Compound 1”	6	7.98 × 10^7^	CuL_2_^+^
“Compound 8”	7.6	2.17 × 10^2^	CuL
“Compound 21”, derivative of 3-methyl-1-(2,4,6-trihydroxy phenyl) butan-1-one	6	2.54 × 10^8^	CuL_2_^−^
“Compound (−)-21a”, derivative of *N*-6-(2-(4-(1*H*-indol-5-yl) piperazin-1-yl)ethyl)-*N*-6-propyl-4,5,6,7-tetrahydrobenzo[*d*]thiazole-2,6-diamine	6	7.98 × 10^7^	CuL_2_^+^
Creatine	6	2.54 × 10^8^	CuL_2_^−^
D512	6	7.98 × 10^7^	CuL_2_^+^
D607(bipyridyl-D2R/D3R agonist hybrid)	6.2	1.87 × 10^4^	CuL
DA-2 (8D)	6.3	8.38 × 10^3^	CuL_2_
DA-3	6	7.98 × 10^7^	CuL_2_
DA-4	6	7.98 × 10^7^	CuL_2_
Dabigatran etexilate	6	7.98 × 10^7^	CuL_2_
Dabrafenib	7.7	1.77 × 10^2^	CuL
2-[[(1,1-Dimethylethyl) oxidoimino]-methyl]-3,5,6-trimethylpyrazine	6.9	1.07 × 10^3^	CuH_2_L_2_
DKP	7.4	3.64 × 10^2^	CuL
Doxycycline	9.2	1.26 × 10^1^	Cu_2_L
Enzastaurin	6	7.98 × 10^7^	CuL_2_
F13714 F15599	6	7.83 × 10^5^	Cu_2_L
Glutathione-hydroxy-quinoline compound	6.3	8.38 × 10^3^	CuL_2_^−^
Glutathione derivatives	15.2	6.21 × 10^−6^	CuHL^−^
Guanabenz	6	7.98 × 10^7^	CuL_2_
8-HQ-MC-5 (VK-28)	6.3	8.38 × 10^3^	CuL_2_
8-hydroxyquinoline	6.3	8.38 × 10^3^	CuL_2_^−^
8-hydroxyquinoline-2-carboxaldehyde isonicotinoyl hydrazone	6.3	8.38 × 10^3^	CuL_2_
Hydroxy-quinoline-propargyl hybrids (HLA20)	6.3	8.38 × 10^3^	CuL_2_
l-(7-Imino-3-propyl-2,3-dihydrothiazolo [4,5-*d*] pyrimidin-6(7*H*)-yl)urea	7.6	2.17 × 10^2^	CuL
KR33493	6	2.54 × 10^8^	CuL_2_
Lestaurtinib	6	7.98 × 10^7^	CuL_2_
M10 M30 (VAR10303) M99	6.3	8.38 × 10^3^	CuL_2_
MAOI-2	6	7.98 × 10^7^	CuL_2_
[18F]MPPF	6	7.98 × 10^7^	CuL_2_
PBF-509	6	7.98 × 10^7^	CuL_2_
PBT2	6.3	8.38 × 10^3^	CuL_2_
Phenothiazine 2Bc (n=0)	6	7.98 × 10^7^	CuL_2_^+^
Piperazine-8-OH-quinolone hybrid	6.3	8.38 × 10^3^	CuL_2_
Preladenant	6	7.98 × 10^7^	CuL_2_
Promethazine	6	7.98 × 10^7^	CuL_2_^+^
Pyrimidinone 8	6	7.98 × 10^7^	CuL_2_
Q1 Q4	6.3	8.38 × 10^3^	CuL_2_
Radotinib	6.2	1.87 × 10^4^	CuL
Rifampicin (ASI-3)	9.2	1.26 × 10^1^	Cu_2_L
Rimonabant	7.4	3.64 × 10^2^	CuL
Rotigotine	7.7	1.77 × 10^2^	CuL
SCH58261 SCH412348	6	7.98 × 10^7^	CuL_2_
ST1535 ST4206	6	7.98 × 10^7^	CuL_2_
Staurosporine	6	7.98 × 10^7^	CuL_2_
V81444	6	2.66 × 10^6^	CuL_2_
VAS3947 VAS2870	7.6	2.17 × 10^2^	CuL
WIN 55, 212-2	6	7.98 × 10^7^	CuL_2_^+^
WR-1065	7.6	2.17 × 10^2^	CuL

**Table 5 biomolecules-09-00269-t005:** pFe(III) and *K*_d_ values, and the most abundant Fe(III) complex, obtained at physiologically relevant conditions: pH = 7.4, *c*_Fe_ = 10^–6^ mol/L, and *c*_L_ = 10^–5^ mol/L. See caption of Table 3 for other notes.

Compound Name(s)	pFe(III)	*K*_d_ (nmol/L)	Most Abundant Complex
7DH 7MH	20.6	2.15 × 10^−1^	FeL_3_
8A 8B 8C	20.6	2.15 × 10^−1^	FeL_3_
8E 8F	21.5	3.01 × 10^−2^	FeH_–2_L_2_
*N*-Acetyl cysteine	16.1	4.59 × 10^9^	FeL_2_^−^
ACPT-I	16.1	9.59 × 10^7^	FeL_2_
Ambroxol	16.3	1.70 × 10^4^	FeL^2+^
Apigenin	16.1	7.55 × 10^8^	FeL^2+^
Apomorphine	16.3	1.35 × 10^4^	FeL_2_
l-Arginine	16.1	1.18 × 10^12^	FeL^3+^
Aromadendrin	16.1	7.55 × 10^8^	FeL
Ascorbic acid	16.1	4.99 × 10^17^	FeL_2_^+^
ASI-1	16.8	2.28 × 10^3^	FeL
ASI-5	18	1.06 × 10^2^	FeL
Astilbin	16.3	1.35 × 10^4^	FeL_2_
Baicalein	16.1	7.55 × 10^8^	FeL^2+^
4*H*-1-benzopyran-4-one	18.1	8.14 × 10^1^	FeL_2_
2,2′-bipyridyl	21.5	3.01 × 10^−2^	FeH_−2_L_2_^+^
4-((5-bromo-3-chloro-2-hydroxybenzyl) amino)-2-hydroxybenzoic acid (LX007, ZL006)	16.1	4.46 × 10^7^	FeL_2_
C-3 (α carboxyfullerene)	16.1	2.54 × 10^10^	FeL_2_
Caffeic acid amide analogues	16.3	1.35 × 10^4^	FeL_2_
Carbidopa	16.2	2.96 × 10^4^	FeL
Carnosic acid	16.3	1.35 × 10^4^	FeL_2_
Catechin	16.1	4.49 × 10^17^	FeHL
Ceftriaxone	16.1	9.59 × 10^7^	FeL_2_
Celastrol	19.2	6.33	FeL_2_
Chebulagic acid	16.1	7.45 × 10^5^	FeHL
Chlorogenic acid	16.1	1.07 × 10^7^	FeL
3′-*O*-(3-Chloropivaloyl) quercetin	16.1	7.55 × 10^8^	FeL
Chrysin	16.1	7.55 × 10^8^	FeL^+^
Clioquinol	20.6	2.15 × 10^−1^	FeL_3_
Clioquinol-selegiline hybrid	22.9	1.07 × 10^−3^	FeL_2_
Clovamide analogues (R_1_ and R_2_ = OH, and/or R_3_ and R_4_ = OH)	16.3	1.35 × 10^4^	FeL_2_
“Compound (−)-8a”	16.3	1.35 × 10^4^	FeL_2_
“Compound 21”, derivative of 3-methyl-1-(2,4,6-trihydroxyphenyl) butan-1-one	16.1	9.59 × 10^7^	FeL_2_^+^
Creatine	16.1	9.59 × 10^7^	FeL_2_^+^
Cudraflavone B	16.1	7.55 × 10^8^	FeL
Curcumin	16.6	4.09 × 10^3^	FeL
Cyanidin	16.3	1.35 × 10^4^	FeL_2_
D607 (bipyridyl-D2R/D3R agonist hybrid)	21.5	3.01 × 10^−2^	FeH_−2_L_2_
DA-2 (8D)	20.6	2.15 × 10^−1^	FeL_3_
DA-3	17.2	5.61 × 10^2^	FeL_2_
DA-4	17.2	5.61 × 10^2^	FeL_2_
Deferasirox	23.5	3.22 × 10^−4^	FeL_2_^3−^
Delphinidin	16.3	1.35 × 10^4^	FeL_2_
Demethoxycurcumin	16.8	2.28 × 10^3^	FeL
Dendropanax morbifera active compound	16.3	1.35 × 10^4^	FeL_2_
Desferrioxamine (Deferoxamine, Desferal, DFO)	26.8	1.81 × 10^−7^	FeHL^+^
4,5-*O*-Dicaffeoyl-1-*O*-(malic acid methyl ester)-quinic acid derivatives (R_1_, R_2_, R_3_, R_4_, or R_5_ = caffeoyl)	16.3	1.35 × 10^4^	FeL_2_
Dihydromyricetin	16.3	1.35 × 10^4^	FeL_2_
5-(3,4-dihydroxybenzylidene)-2,2-dimethyl-1,3-dioxane-4,6-dione	16.3	1.35 × 10^4^	FeL_2_^−^
7,8-dihydroxycoumarin derivative DHC12	16.3	1.35 × 10^4^	FeL_2_
3′,4′-dihydroxyflavone	16.3	1.35 × 10^4^	FeL_2_^−^
7,8-dihydroxyflavone	16.3	1.35 × 10^4^	FeL_2_^−^
5,7-dihydroxy-4′-methoxyflavone	18.1	8.14 × 10^1^	FeL_2_
(*E*)-3,4-dihydroxystyryl aralkyl sulfones	16.3	1.35 × 10^4^	FeL_2_^−^
(*E*)-3,4-dihydroxystyryl aralkyl sulfoxides	16.3	1.35 × 10^4^	FeL_2_^−^
5,3′-dihydroxy-3,7,4′-trimethoxyflavone	18.1	8.14 × 10^1^	FeL_2_
2-[[(1,1-Dimethylethyl) oxidoimino]-methyl]-3,5,6-trimethylpyrazine	16.1	1.86 × 10^10^	FeL
DKP	16.1	7.43 × 10^21^	FeL_2_
l-DOPA (levodopa, CVT-301)	16.2	2.96 × 10^4^	FeL
DOPA-derived peptido-mimetics (deprotected)	16.2	2.96 × 10^4^	FeL
DOPA-derived peptido-mimetics (protected)	16.3	1.35 × 10^4^	FeL_2_
l-dopa deuterated	16.2	2.96 × 10^4^	FeL
Doxycycline	18.1	7.22 × 10^1^	FeL_2_
Droxidopa	16.2	2.96 × 10^4^	FeL
Echinacoside	16.3	1.35 × 10^4^	FeL_2_
Ellagic acid	16.3	1.35 × 10^4^	FeL_2_
Entacapone (comtan, ASI-6)	19.3	3.99	FeL_3_^3−^
Epicatechin	16.3	1.35 × 10^4^	FeL_2_
Epigallocatechin-3-gallate	16.1	7.45 × 10^5^	FeHL
Etidronate (HEDPA)	23.5	3.22 × 10^−4^	FeH_−1_L
Exifone	16.1	7.45 × 10^5^	FeHL
Farrerol	16.1	7.55 × 10^8^	FeL
Fiset (3,3′,4′,7-tetra-hydroxy-flavone)	16.3	1.35 × 10^4^	FeL_2_
Fraxetin	16.3	1.35 × 10^4^	FeL_2_^−^
Galangin	27.0	9.36 × 10^−8^	FeH_−1_L
Gallic acid derivatives	16.1	7.45 × 10^5^	FeHL
Garcinol	16.3	1.35 × 10^4^	FeL_2_
Genistein	16.1	7.55 × 10^8^	FeL
Glutathione-hydroxy-quinoline compound	18	1.04 × 10^2^	FeH_−2_L^+^
Glutathione-l-DOPA compound	16.3	1.35 × 10^4^	FeL_2_
Gly-N-C-DOPA	16.2	2.96 × 10^4^	FeL
Hesperidin	16.1	7.55 × 10^8^	FeL
Hinokitiol	16.1	4.30 × 10^8^	FeL^2+^
8-HQ-MC-5 (VK-28)	20.6	2.15 × 10^−1^	FeL_3_
4-Hydroxyisophthalic acid	16.1	1.64 × 10^6^	FeL_3_
1-Hydroxy-2-pyridinone derivatives	17.7	1.50 × 10^2^	FeL_3_
3-Hydroxy-4(1*H*)pyridinone (Deferiprone)	19.3	3.92	FeL_3_
3-Hydroxy-4(1*H*)pyridinone derivatives (R = H)	19.3	3.92	FeL_3_
8-Hydroxyquinoline	20.6	2.15 × 10^−1^	FeL_3_
8-Hydroxyquinoline-2-carboxaldehyde isonicotinoyl hydrazone	20.6	2.15 × 10^−1^	FeL_3_
Hydroxy-quinoline-propargyl hybrids (HLA20)	20.6	2.15 × 10^−1^	FeL_3_
Hydroxytyrosol butyrate	16.3	1.35 × 10^4^	FeL_2_^−^
Hyperoside	27.0	9.36 × 10^−8^	FeH_−1_L
Icariin	18.1	8.14 × 10^1^	FeL_2_
Icariside II	18.1	8.14 × 10^1^	FeL_2_
Isobavachalcone	16.1	3.23 × 10^8^	FeL_2_^+^
Isochlorogenic acid	16.1	1.07 × 10^7^	FeL
Isoquercetin (isoquercitrin)	27.0	9.36 × 10^−8^	FeH_−1_L
Kaempferol	27.0	9.36 × 10^−8^	FeH_−1_L
Kaempferol, 3-*O*-a-L arabino-furanoside-7-*O*-a-L- rhamno-pyranoside	18.1	8.14 × 10^1^	FeL_2_
KR33493	16.3	1.06 × 10^4^	FeL_2_
Kukoamine	16.3	1.35 × 10^4^	FeL_2_
Luteolin	16.3	1.35 × 10^4^	FeL_2_
M10 M30 (VAR10303) M99	20.6	2.15 × 10^−1^	FeL_3_
Macranthoin G	16.3	1.35 × 10^4^	FeL_2_
Magnesium lithospermate B	16.3	1.35 × 10^4^	FeL_2_
α-mangostin	16.3	1.35 × 10^4^	FeL_2_
γ-mangostin	16.3	1.35 × 10^4^	FeL_2_
Metformin (Met)	16.1	1.36 × 10^9^	FeL_2_^+^
MitoQ	16.1	5.72 × 10^10^	FeL_2_
Morin	18.1	8.14 × 10^1^	FeL_2_
Myricetin Myricitrin	27.0	9.36 × 10^−8^	FeH_−1_L
Naringenin	16.1	7.55 × 10^8^	FeL
Naringin	16.1	7.55 × 10^8^	FeL
Nicotinamide adenine dinucleotide phosphate (NADPH)	16.1	6.01 × 10^10^	FeL_2_
Nitecapone	16.8	2.04 × 10^3^	FeL_2_^−^
Nordihydroguaiaretic acid	16.3	1.35 × 10^4^	FeL_2_
Oleuropein	16.3	1.35 × 10^4^	FeL_2_^−^
Opicapone	16.8	2.04 × 10^3^	FeL_2_
PBT2	20.6	2.15 × 10^−1^	FeL_3_
PBT434	16.1	3.22 × 10^9^	FeL^2+^
Petunidin	16.3	1.35 × 10^4^	FeL_2_
Phenylhydroxamates	16.1	9.46 × 10^4^	FeL_2_
Piceatannol	16.3	1.35 × 10^4^	FeL_2_
Pinostrobin (5-hydroxy-7-methoxy-flavone)	18.1	8.14 × 10^1^	FeL_2_
Piperazine-8-OH-quinolone hybrid	20.6	2.15 × 10^−1^	FeL_3_
Protocatechuic acid	22.2	6.16 × 10^−3^	FeL_2_^3−^
Protosappanin A	16.3	1.35 × 10^4^	FeL_2_
Punicalangin	16.1	1.14 × 10^12^	FeL
Pyridoxal isonicotinoyl hydrazone (PIH)	22.9	1.07 × 10^−3^	FeL_2_
Pyridoxal isonicotinoyl hydrazone derivatives: PCIH PCTH H2NPH H2PPH	22.9	1.07 × 10^−3^	FeL_2_
Q1 Q4	20.6	2.15 × 10^−1^	FeL_3_
Quercetin	27.0	9.36 × 10^−8^	FeH_–1_L^3−^
Quinoline derivatives as SUMOylation activators	16.1	1.14 × 10^16^	FeL^2+^
Radotinib	21.5	3.01 × 10^−2^	FeH_−2_L_2_
Rimonabant	16.1	7.43 × 10^21^	FeL_2_
Rosmarinic acid	16.3	1.35 × 10^4^	FeL_2_
Rutin	27.0	9.36 × 10^−8^	FeH_−1_L
Salicylate, sodium salt	16.1	1.64 × 10^6^	FeL_3_^3−^
Salvianolic acid B	16.3	1.35 × 10^4^	FeL_2_
Silibinin (silybin) A, B	16.1	1.99 × 10^19^	FeH_3_L^3+^
Silydianin	16.1	1.99 × 10^19^	FeH_3_L^3+^
Sulfuretin	16.3	1.35 × 10^4^	FeL_2_
Tanshinol	16.3	1.35 × 10^4^	FeL_2_
Tannic acid	16.1	6.05 × 10^49^	Fe_4_L
Taxifolin	16.1	7.55 × 10^8^	FeL
Tectorigenin	16.1	7.55 × 10^8^	FeL
Tetracycline	16.1	8.35 × 10^10^	FeL_2_^2−^
Tolcapone (ASI-7)	16.8	2.04 × 10^3^	FeL_2_^−^
Transilitin	16.3	1.35 × 10^4^	FeL_2_
*o*-Trensox	29.5	3.36 × 10^−10^	FeL^3−^
2,3,3-Trisphosphonate	18	1.06 × 10^2^	FeL
Verbascoside	16.3	1.35 × 10^4^	FeL_2_
Zonisamide	16.1	1.86 × 10^11^	FeL_2_

**Table 6 biomolecules-09-00269-t006:** pFe(II) and *K*_d_ values, and the most abundant Fe(II) complex, obtained at physiologically relevant conditions: pH = 7.4, *c*_Fe_ = 10^−6^ mol/L, and *c*_L_ = 10^−5^ mol/L. See caption of Table 3 for other notes.

Compound Name(s)	pFe(II)	*K*_d_ (nmol/L)	Most Abundant Complex
7DH 7MH	6.9	1.35 × 10^3^	FeL
8A 8B 8C	6.9	1.35 × 10^3^	FeL
8E 8F	6.1	4.62 × 10^4^	FeL_2_
ACPT-I	6	1.20 × 10^7^	FeL
Alvespimycin	6	2.07 × 10^7^	FeL^2+^
Aminothiazoles derivatives as SUMOylation activators	6	2.77 × 10^6^	FeL
AMN082	6	2.07 × 10^7^	FeL^2+^
Apomorphine	6	1.53 × 10^7^	FeHL
l-Arginine	6	3.30 × 10^7^	FeL^2+^
Ascorbic acid	6	1.89 × 10^9^	FeL^+^
ASI-1	6	1.78 × 10^5^	FeL^+^
ASI-5	10.7	1.95 × 10^−1^	FeL
Astilbin	6	1.53 × 10^7^	FeHL
Azilsartan	6	2.53 × 10^15^	FeL_2_
Baicalein	9.9	1.09	FeL_2_^2−^
Benserazide	9.9	1.09	FeL_2_
(−)-*N*6-(2-(4-(Biphenyl-4-yl)piperazin-1-yl)-ethyl)-*N*6-propyl-4,5,6,7-tetrahydrobenzo[*d*]thiazole-2,6-diamine derivatives	6	2.07 × 10^7^	FeL^2+^
2,2′-Bipyridyl	6.1	4.62 × 10^4^	FeL_2_^2+^
4-((5-bromo-3-chloro-2-hydroxybenzyl) amino)-2-hydroxybenzoic acid (LX007, ZL006)	6	3.13 × 10^8^	FeL
C-3 (α carboxyfullerene)	6	6.90 × 10^6^	FeL
Caffeic acid amide analogues	6.7	2.02 × 10^3^	FeH_−1_L
Carbazole-derived compounds	6	2.07 × 10^7^	FeL^2+^
Carbidopa	6.3	1.07 × 10^4^	FeL
Carnosic acid	6	1.53 × 10^7^	FeHL
Catechin	6	1.53 × 10^7^	FeHL
Ceftriaxone	6	1.20 × 10^7^	FeL
Celastrol	6	1.53 × 10^7^	FeHL
CEP-1347	6	1.53 × 10^7^	FeHL
Chebulagic acid	6	5.66 × 10^5^	FeL
Chlorogenic acid	6	2.93 × 10^6^	FeHL
Clioquinol	7.9	1.01 × 10^2^	FeL_2_
Clioquinol-selegiline hybrid	7.1	6.37 × 10^2^	FeH_2_L_2_
Clovamide analogues (R_1_ and R_2_ = OH, and/or R_3_ and R_4_ = OH)	6	1.53 × 10^7^	FeHL
“Compound 1”	6	2.07 × 10^7^	FeL^2+^
“Compound (−)-8a”	6	1.53 × 10^7^	FeHL
“Compound 21”, derivative of 3-methyl-1-(2,4,6-trihydroxyphenyl) butan-1-one	6	1.20 × 10^7^	FeL^+^
“Compound (−)-21a”, derivative of *N*-6-(2-(4-(1*H*-indol-5-yl) piperazin-1-yl)ethyl)-*N*-6-propyl-4,5,6,7-tetrahydrobenzo[*d*]thiazole-2,6-diamine	6	2.07 × 10^7^	FeL^2+^
Creatine	6	1.20 × 10^7^	FeL^+^
Curcumin	6	4.16 × 10^9^	FeH_2_L^+^
Cyanidin	6	1.53 × 10^7^	FeHL
D512	6	2.07 × 10^7^	FeL^2+^
D607 (bipyridyl-D2R/D3R agonist hybrid)	6.1	4.63 × 10^4^	FeL_2_
DA-2 (8D)	6.9	1.35 × 10^3^	FeL
DA-3	6	2.07 × 10^7^	FeL
DA-4	6	2.07 × 10^7^	FeL
Dabigatran etexilate	6	2.07 × 10^7^	FeL
Delphinidin	9.9	1.09	FeL_2_
Demethoxycurcumin	6	1.78 × 10^5^	FeL
Dendropanax morbifera	6	1.53 × 10^7^	FeHL
Desferrioxamine (Deferoxamine, Desferal, DFO)	6.2	2.24 × 10^4^	FeH_2_L^+^
4,5-*O*-Dicaffeoyl-1-*O*-(malic acid methyl ester)-quinic acid derivatives (R_1_, R_2_, R_3_, R_4_, or R_5_ = caffeoyl)	6	1.53 × 10^7^	FeHL
Dihydromyricetin	9.9	1.09	FeL_2_
5-(3,4-Dihydroxybenzylidene)-2,2-dimethyl-1,3-dioxane-4,6-dione	6	1.53 × 10^7^	FeHL^+^
7, 8-Dihydroxycoumarin derivative DHC12	6	1.53 × 10^7^	FeHL
3′,4′-Dihydroxyflavone	14.8	1.58 × 10^−5^	FeL^+^
7,8-Dihydroxyflavone	6	1.53 × 10^7^	FeHL^+^
(*E*)-3,4-Dihydroxystyryl aralkyl sulfones	6	1.53 × 10^7^	FeHL^+^
(*E*)-3,4-Dihydroxystyryl aralkyl sulfoxides	6	1.53 × 10^7^	FeHL^+^
2-[[(1,1-Dimethylethyl) oxidoimino]-methyl]-3,5,6-trimethylpyrazine	8.2	5.40 × 10^1^	FeL_2_
DKP	6	4.33 × 10^5^	FeL
l-DOPA (levodopa, CVT-301)	6.3	1.07 × 10^4^	FeL^−^
DOPA-derived peptido-mimetics (deprotected)	10.5	2.82 × 10^−1^	FeHL
DOPA-derived peptido-mimetics (protected)	6	1.53 × 10^7^	FeHL
l-DOPA deuterated	6.3	1.07 × 10^4^	FeL^−^
Doxycycline	6	4.07 × 10^8^	FeL
Droxidopa	6.3	1.07 × 10^4^	FeL^−^
Echinacoside	6	1.53 × 10^7^	FeHL
Ellagic acid	6	1.53 × 10^7^	FeHL
Entacapone (comtan, ASI-6)	12.7	1.65 × 10^−3^	FeL_2_^2−^
Enzastaurin	6	2.07 × 10^7^	FeL
Epicatechin	6	1.53 × 10^7^	FeHL
Etidronate (HEDPA)	9.9	1.14	FeL^2−^
F13714 F15599	6	2.77 × 10^6^	FeL
FIsetin (3,3′,4′,7-tetra-hydroxy-flavone)	6	1.53 × 10^7^	FeHL
Fraxetin	6	1.53 × 10^7^	FeHL^+^
Gallocatechin	9.9	1.09	FeL_2_
Garcinol	6	1.53 × 10^7^	FeHL
Glutamine	6	7.54 × 10^5^	FeL^+^
Glutathione-hydroxy-quinoline compound	6.9	1.35 × 10^3^	FeL^+^
Glutathione-l-DOPA compound	6	1.53 × 10^7^	FeHL
Gly-N-C-DOPA	6.3	1.07 × 10^4^	FeL^−^
Guanabenz	6	2.07 × 10^7^	FeL
8-HQ-MC-5 (VK28)	6.9	1.35 × 10^3^	FeL
4-Hydroxyisophthalic acid	6	3.13 × 10^8^	FeL
8-hydroxyquinoline	6.9	1.35 × 10^3^	FeL^+^
8-hydroxyquinoline-2-carboxaldehyde isonicotinoyl hydrazone	6.9	1.35 × 10^3^	FeL
Hydroxy-quinoline-propargyl hybrids (HLA20)	6.9	1.35 × 10^3^	FeL
Hydroxytyrosol butyrate	6	1.53 × 10^7^	FeHL^+^
Isochlorogenic acid	6	2.93 × 10^6^	FeHL
KR33493	6	1.20 × 10^7^	FeL
Kukoamine	6	1.53 × 10^7^	FeHL
Lestaurtinib	6	2.07 × 10^7^	FeL
M10 M30 (VAR10303) M99	6.9	1.35 × 10^3^	FeL
Macranthoin G	6	1.53 × 10^7^	FeHL
Magnesium lithospermate B	6	1.53 × 10^7^	FeHL
*α*-mangostin	6	1.53 × 10^7^	FeHL
*γ*-mangostin	6	1.53 × 10^7^	FeHL
MAOI-1	9.9	1.09	FeL_2_
MAOI-2	6	2.07 × 10^7^	FeL
Meclofenamic acid	6	2.53 × 10^15^	FeL_2_
Mildronate	6	2.53 × 10^15^	FeL_2_
Mitomycin C	6	6.57 × 10^7^	FeL
[18F]MPPF	6	2.07 × 10^7^	FeL
Nitecapone	12.7	1.65 × 10^−3^	FeL_2_^2−^
Nordihydroguaiaretic acid	6	1.53 × 10^7^	FeHL
Oleuropein	6	1.53 × 10^7^	FeHL^+^
Opicapone	12.7	1.65 × 10^−3^	FeL_2_
PBF-509	6	2.07 × 10^7^	FeL
PBT2	6.9	1.35 × 10^3^	FeL
PBT434	6.2	1.80 × 10^4^	FeL^+^
Petunidin	6	1.53 × 10^7^	FeHL
Phenothiazine 2Bc (n=0)	6	2.07 × 10^7^	FeL^2+^
Phenylhydroxamates	6	5.96 × 10^7^	FeL_2_
Piceatannol	6	1.53 × 10^7^	FeHL
Piperazine-8-OH-quinolone hybrid	6.9	1.35 × 10^3^	FeL
Preladenant	6	2.07 × 10^7^	FeL
Promethazine	6	2.07 × 10^7^	FeL^2+^
Protosappanin A	6	1.53 × 10^7^	FeHL
Pyridoxal isonicotinoyl hydrazone (PIH)	7.1	6.37 × 10^2^	FeH_2_L_2_
Pyridoxal isonicotinoyl hydrazone derivatives: PCIH PCTH H2NPH H2PPH	7.1	6.37 × 10^2^	FeH_2_L_2_^2+^
Pyrimidinone 8	6	2.07 × 10^7^	FeL
Q1 Q4	6.9	1.35 × 10^3^	FeL
Radotinib	6.1	4.62 × 10^4^	FeL_2_
Riboflavin	6	1.22 × 10^5^	FeL^2+^
Rifampicin (ASI-3)	6	4.07 × 10^8^	FeL
Rimonabant	6	4.85 × 10^11^	FeL
Rosmarinic acid	6	1.53 × 10^7^	FeLH
Salicylate, sodium salt	6	3.13 × 10^8^	FeL
Salvianolic acid B	6	1.53 × 10^7^	FeHL
SCH58261SCH412348	6	2.07 × 10^7^	FeL
ST1535 ST4206	6	2.07 × 10^7^	FeL
Staurosporine	6	2.07 × 10^7^	FeL
Sulfuretin	6	1.53 × 10^7^	FeHL
Tanshinol	6	1.53 × 10^7^	FeHL
Tetracycline	6	2.73 × 10^6^	FeL
Tolcapone (ASI-7)	12.7	1.65 × 10^−3^	FeL_2_^2−^
Transilitin	6	1.53 × 10^7^	FeHL
2′,3′,4′-trihydroxyflavone	9.9	1.09	FeL_2_^2−^
2,3,3-trisphosphonate	10.7	1.95 × 10^−1^	FeL
V81444	6	2.77 × 10^6^	FeL
Verbascoside	6	2.02 × 10^6^	FeL
WIN 55, 212-2	6	2.07 × 10^7^	FeL^2+^

**Table 7 biomolecules-09-00269-t007:** pMn(II) and *K*_d_ values, and the most abundant Mn(II) complex, obtained at physiologically relevant conditions: pH = 7.4, *c*_Mn_ = 10^−6^ mol/L, and *c*_L_ = 10^−5^ mol/L. See caption of Table 3 for other notes.

Compound Name(s)	pMn(II)	*K*_d_ (nmol/L)	Most Abundant Complex
7DH 7MH	6.7	2.35 × 10^3^	MnL^+^
8A 8B 8C	6.7	2.35 × 10^3^	MnL
8E 8F	6	2.40 × 10^6^	MnL
*N*-Acetyl cysteine	6	8.97 × 10^5^	MnHL^+^
ACPT-I	6	2.25 × 10^8^	MnL
Alvespimycin	6	7.61 × 10^8^	MnL^2+^
Ambroxol	6.7	2.47 × 10^3^	MnL^+^
3-(7-amino-5-(cyclohexylamino)-[1,2,4]triazolo[1,5-*a*][1,3,5]triazin-2-yl)-2-cyanoacrylamide	6	7.61 × 10^8^	MnL
Aminothiazoles derivatives as SUMOylation activators	6	7.64 × 10^7^	MnL
AMN082	6	7.61 × 10^8^	MnL^2+^
Apomorphine	6	7.16 × 10^8^	MnL
l-Arginine	6	1.45 × 10^8^	MnL^2+^
ASI-1	6	3.44 × 10^6^	MnL^+^
ASI-5	6	7.81 × 10^6^	MnL
Astilbin	6	7.16 × 10^8^	MnL
Azilsartan	6	3.71 × 10^9^	MnL
Baicalein	6	2.62 × 10^7^	MnL
Benserazide	6	2.62 × 10^7^	MnL
(−)-*N*6-(2-(4-(Biphenyl-4-yl)piperazin-1-yl)-ethyl)-*N*6-propyl-4,5,6,7-tetrahydrobenzo[*d*]thiazole-2,6-diamine derivatives	6	7.61 × 10^8^	MnL^2+^
2,2′-bipyridyl	6	2.40 × 10^6^	MnL^2+^
4-((5-bromo-3-chloro-2-hydroxybenzyl) amino)-2-hydroxybenzoic acid (LX007, ZL006)	6	8.72 × 10^8^	MnL
C-3 (α carboxyfullerene)	6	5.06 × 10^6^	MnL
Caffeic acid amide analogues	6	6.62 × 10^7^	MnH_−1_L
Carbazole-derived compounds	6	7.61 × 10^8^	MnL^2+^
Carbidopa	7.6	2.33 × 10^2^	MnHL
Carnosic acid	6	7.16 × 10^8^	MnL
Catechin	6	7.16 × 10^8^	MnL
Ceftriaxone	6	2.25 × 10^8^	MnL
Celastrol	6	7.16 × 10^8^	MnL
CEP1347	6	3.71 × 10^9^	MnL
Chebulagic acid	6	2.62 × 10^7^	MnL
Chlorogenic acid	6	3.91 × 10^7^	MnL^−^
Clioquinol	6.7	2.35 × 10^3^	MnL^+^
Clovamide analogues (R_1_ and R_2_ = OH, and/or R_3_ and R_4_ = OH)	6	7.16 × 10^8^	MnL
“Compound 1”	6	7.61 × 10^8^	MnL^++^
“Compound (−)-8a”	6	7.16 × 10^8^	MnL
“Compound 21”, derivative of 3-methyl-1-(2,4,6-trihydroxyphenyl) butan-1-one	6	2.25 × 10^8^	MnL^+^
“Compound (−)-21a”, derivative of *N*-6-(2-(4-(1*H* -indol-5-yl)piperazin-1-yl)ethyl)-*N*-6-propyl-4,5,6,7-tetrahydrobenzo[*d*]thiazole-2,6-diamine	6	7.61 × 10^8^	MnL^++^
Creatine	6	2.25 × 10^8^	MnL^+^
Curcumin	6	3.44 × 10^6^	MnL
Cyanidin	6	7.16 × 10^8^	MnL
D512	6	7.61 × 10^8^	MnL^2+^
D607 (bipyridyl-D2R/D3R agonist hybrid)	6	2.40 × 10^6^	MnL
DA-2 (8D)	6.7	2.35 × 10^3^	MnL
DA-3	6	7.61 × 10^8^	MnL
DA-4	6	7.61 × 10^8^	MnL
Dabigatran etexilate	6	7.61 × 10^8^	MnL
(*S*)-3,4-DCPG	6	5.91 × 10^6^	MnL
Delphinidin	6	2.62 × 10^7^	MnL
Demethoxycurcumin	6	3.44 × 10^6^	MnL
Dendropanax morbifera active compound	6	7.16 × 10^8^	MnL
4,5-*O*-Dicaffeoyl-1-*O*-(malic acid methyl ester)-quinic acid derivatives (R_1_, R_2_, R_3_, R_4_, or R_5_ = caffeoyl)	6	7.16 × 10^8^	MnL
Dihydromyricetin	6	2.62 × 10^7^	MnL
5-(3,4-Dihydroxybenzylidene) -2,2-dimethyl-1,3-dioxane-4,6-dione	6	7.16 × 10^8^	MnL
7,8-Dihydroxycoumarin derivative DHC12	6	7.16 × 10^8^	MnL
3′,4′-Dihydroxyflavone	6	7.16 × 10^8^	MnL
7,8-dihydroxyflavone	6	7.16 × 10^8^	MnL
(*E*)-3,4-Dihydroxystyryl aralkyl sulfones	6	7.16 × 10^8^	MnL
(*E*)-3,4-Dihydroxystyryl aralkyl sulfoxides	6	7.16 × 10^8^	MnL
2-[[(1,1-Dimethylethyl) oxidoimino]-methyl]-3,5,6-trimethylpyrazine	6	2.58 × 10^6^	MnL
DKP	6	1.00 × 10^6^	MnHL
l-DOPA (levodopa, CVT-301)	7.6	2.26 × 10^2^	MnHL
DOPA-derived peptido-mimetics (deprotected)	7.6	2.33 × 10^2^	MnHL
DOPA-derived peptido-mimetics (protected)	6	7.16 × 10^8^	MnL
l-dopa deuterated	7.6	2.26 × 10^2^	MnHL
Droxidopa	7.6	2.33 × 10^2^	MnHL
Echinacoside	6	7.16 × 10^8^	MnL
Ellagic acid	6	7.16 × 10^8^	MnL
Entacapone (comtan, ASI-6)	6	6.74 × 10^5^	MnL
Enzastaurin	6	7.61 × 10^8^	MnL
Epicatechin	6	7.16 × 10^8^	MnL
Etidronate (HEDPA)	6	1.03 × 10^6^	MnL^2−^
F13714 F15599	6	7.64 × 10^7^	MnL
Fisetin (3,3′,4′,7-tetra-hydroxy-flavone)	6	7.16 × 10^8^	MnL
Fraxetin	6	7.16 × 10^8^	MnL
Gallocatechin	6	2.62 × 10^7^	MnL
Garcinol	6	7.16 × 10^8^	MnL
Glutamine	6	2.30 × 10^7^	MnL^+^
Glutathione derivatives	6	4.07 × 10^5^	MnL^−^
Glutathione-hydroxy-quinoline compound	6.7	2.35 × 10^3^	MnL^+^
Glutathione-l-DOPA compound	6	7.16 × 10^8^	MnL
Gly-N-C-DOPA	7.6	2.33 × 10^2^	MnHL
Guanabenz	6	7.61 × 10^8^	MnL
Hinokitiol	6.1	2.99 × 10^4^	MnL^+^
8-HQ-MC-5 (VK-28)	6.7	2.35 × 10^3^	MnL
4-Hydroxyisophthalic acid	6	8.72 × 10^8^	MnL
8-hydroxyquinoline	6.7	2.35 × 10^3^	MnL^+^
8-Hydroxyquinoline-2-carboxaldehyde isonicotinoyl hydrazone	6.7	2.35 × 10^3^	MnL
Hydroxy-quinoline-propargyl hybrids (HLA20)	6.7	2.35 × 10^3^	MnL
Hydroxytyrosol butyrate	6	7.16 × 10^8^	MnL
Isobavachalcone	6.2	1.55 × 10^4^	MnL^+^
Isochlorogenic acid	6	3.91 × 10^7^	MnL^−^
KR33493	6	2.25 × 10^8^	MnL
Kukoamine	6	7.16 × 10^8^	MnL
Lestaurtinib	6	7.61 × 10^8^	MnL
Lipoic acid	6	9.53 × 10^6^	MnL^+^
Luteolin	6	7.16 × 10^8^	MnL
M10 M30 (VAR10303) M99	6.7	2.35 × 10^3^	MnL
Macranthoin G	6	7.16 × 10^8^	MnL
Magnesium lithospermate B	6	7.16 × 10^8^	MnL
α-mangostin	6	7.16 × 10^8^	MnL
γ-mangostin	6	7.16 × 10^8^	MnL
MAOI-1	6	2.62 × 10^7^	MnL
MAOI-2	6	7.61 × 10^8^	MnL
Meclofenamic acid	6	3.71 × 10^9^	MnL
Mildronate	6	3.71 × 10^9^	MnL
Mitomycin C	6	7.46 × 10^7^	MnL
MitoQ	6	1.29 × 10^5^	MnL
[18F]MPPF	6	7.61 × 10^8^	MnL
Nicotinamide adenine dinucleotide phosphate (NADPH)	6	2.01 × 10^7^	MnL
Nicotinamide mononucleotide	6	6.59 × 10^6^	MnL
Nitecapone	6	6.74 × 10^5^	MnL
Nordihydroguaiaretic acid	6	7.16 × 10^8^	MnL
Oleuropein	6	7.16 × 10^8^	MnL
Opicapone	6	6.74 × 10^5^	MnL
PBF-509	6	2.41 × 10^9^	MnL
PBT2	6	5.22 × 10^5^	MnL^+^
Petunidin	6	7.16 × 10^8^	MnL
Phenothiazine 2Bc (n=0)	6	7.61 × 10^8^	MnL^2+^
Phenylhydroxamates	6	5.81 × 10^6^	MnL
Piceatannol	6	7.16 × 10^8^	MnL
Piperazine-8-OH-quinolone hybrid	6.7	2.35 × 10^3^	MnL
Preladenant	6	7.61 × 10^8^	MnL
Promethazine	6	7.61 × 10^8^	MnL^2+^
Protocatechuic acid	6	3.66 × 10^8^	MnL^−^
Protosappanin A	6	7.16 × 10^8^	MnL
Pyrimidinone 8	6	7.61 × 10^8^	MnL
Q1 Q4	6.7	2.35 × 10^3^	MnL
Radotinib	6	2.40 × 10^6^	MnL
Riboflavin	6	5.75 × 10^5^	MnHL^3+^
Rifampicin (ASI-3)	6	1.03 × 10^6^	MnL
Rimonabant	6	1.00 × 10^6^	MnHL
Rosmarinic acid	6	7.16 × 10^8^	MnL
Salicylate, sodium salt	6	8.72 × 10^8^	MnL
Salvianolic acid B	6	7.16 × 10^8^	MnL
SCH58261 SCH412348	6	7.61 × 10^8^	MnL
ST1535 ST4206	6	7.61 × 10^8^	MnL
Staurosporine	6	7.61 × 10^8^	MnL
Sulfuretin	6	7.16 × 10^8^	MnL
Tanshinol	6	7.16 × 10^8^	MnL
Taurine	6	6.39 × 10^11^	MnL_2_
Tetracycline	6	2.14 × 10^7^	MnL
Tolcapone (ASI-7)	6	6.74 × 10^5^	MnL
Transilitin	6	7.16 × 10^8^	MnL
2′,3′,4′-Trihydroxyflavone	6	2.62 × 10^7^	MnL
V81444	6	7.64 × 10^7^	MnL
Verbascoside	6	6.62 × 10^7^	MnH_−1_L
WIN 55,212-2	6	7.61 × 10^8^	MnL^2+^

**Table 8 biomolecules-09-00269-t008:** pZn(II) and *K*_d_ values, and the most abundant Zn(II) complex, obtained at physiologically relevant conditions: pH = 7.4, *c*_Zn_ = 10^−6^ mol/L, and *c*_L_ = 10^−5^ mol/L. See caption of Table 3 for other notes.

Compound Name(s)	pZn(II)	*K*_d_ (nmol/L)	Most Abundant Complex
7DH 7MH	7.5	3.16 × 10^2^	ZnL
8A 8B 8C	7.5	3.16 × 10^2^	ZnL
8E 8F	6.4	6.31 × 10^3^	ZnL
*N*-acetyl cystein	6	1.66 × 10^6^	ZnL
ACPT-I	6	1.46 × 10^7^	ZnL
Alaternin	6.8	1.83 × 10^3^	ZnL
Alvespimycin	6	7.85 × 10^5^	ZnL^2+^
AM-251	6	1.95 × 10^6^	ZnHL
Ambroxol	7.7	1.90 × 10^2^	ZnL_2_
3-(7-amino-5-(cyclohexylamino)-[1,2,4]triazolo[1,5-*a*][1,3,5]triazin-2-yl)-2-cyanoacrylamide	6	7.85 × 10^5^	ZnL
Aminothiazoles derivatives as SUMOylation activators	6	2.28 × 10^5^	ZnL
AMN082	6	7.85 × 10^5^	ZnL^2+^
Antagonist of the A(2A) adenosine receptor - derivative 49	6	1.95 × 10^6^	ZnHL
Apigenin	6	1.23 × 10^29^	ZnH_3_L
Apomorphine	6	7.76 × 10^6^	ZnL
l-Arginine	6	4.14 × 10^6^	ZnL^2+^
Ascorbic acid	6.1	4.67 × 10^4^	ZnL^+^
ASI-1	6	5.76 × 10^5^	ZnL^+^
ASI-5	6	9.06 × 10^6^	ZnL
Astilbin	6	7.76 × 10^6^	ZnL
Azilsartan	6	5.22 × 10^5^	ZnL
Baicalein	6	2.71 × 10^5^	ZnL
Benserazide	6	2.71 × 10^5^	ZnL
7*H*-Benzo[*e*] perimidin-7-one derivatives (R_6_ = OH)	6.1	4.82 × 10^4^	ZnL_2_
4*H*-1-benzopyran-4-one	6	3.56 × 10^20^	ZnH_3_L
8-Benzyl-tetrahydropyrazino[2,1-*f*]purinedione (derivative n. 57)	6	9.67 × 10^11^	ZnL
Bikaverin	6.3	1.07 × 10^4^	ZnH_−2_L^2−^
(−)-*N*6-(2-(4-(Biphenyl-4-yl)piperazin-1-yl)-ethyl)-*N*6-propyl-4,5,6,7-tetrahydrobenzo[d]thiazole-2,6-diamine derivatives	6	7.85 × 10^5^	ZnL^2+^
2,2′-bipyridyl	6.4	6.31 × 10^3^	ZnL^2+^
4-((5-Bromo-3-chloro-2-hydroxybenzyl) amino)-2-hydroxybenzoic acid (LX007, ZL006)	6	8.04 × 10^8^	ZnH_−1_L
C-3 (α carboxyfullerene)	6	2.08 × 10^6^	ZnL
Caffeic acid amide analogues	6	9.29 × 10^5^	ZnH_−1_L
Carbazole-derived compounds	6	7.85 × 10^5^	ZnL^2+^
Carbidopa	6	2.96 × 10^6^	ZnHL
Carnosic acid	6	7.76 × 10^6^	ZnL
Cathechin	6	7.76 × 10^6^	ZnL
Ceftriaxone	6.1	3.68 × 10^4^	ZnL
Celastrol	6	7.76 × 10^6^	ZnL
Chebulagic acid	6	1.81 × 10^13^	Zn_2_L
Chlorogenic acid	6	6.86 × 10^5^	ZnL^−^
3′-*O*-(3-Chloropivaloyl) quercetin	6	5.69 × 10^16^	ZnH_3_L
Chlorpromazine	6	1.95 × 10^6^	ZnHL^3+^
Chrysin	6	5.69 × 10^16^	ZnH_3_L
Clioquinol	7.5	3.16 × 10^2^	ZnL^+^
Clovamide analogues (R_1_ and R_2_ = OH, and/or R_3_ and R_4_ = OH)	6	7.76 × 10^6^	ZnL
“Compound 1”	6	7.85 × 10^5^	ZnL^2+^
“Compound (−)-8a”	6	7.76 × 10^6^	ZnL
“Compound 8”	6.4	6.47 × 10^3^	ZnH_−1_L
“Compound 21”, derivative of 3-methyl-1-(2,4,6-trihydroxyphenyl) butan-1-one	6	1.54 × 10^6^	ZnL^+^
“Compound (−)-21a”, derivative of *N*-6-(2-(4-(1*H* -indol-5-yl)piperazin-1-yl)ethyl)-*N*-6-propyl-4,5,6,7-tetrahydrobenzo [*d*]thiazole-2,6-diamine	6	7.85 × 10^5^	ZnL^2+^
Creatine	6	1.54 × 10^6^	ZnL^+^
Cudraflavone B	6	5.69 × 10^16^	ZnH_3_L
Curcumin	6	5.76 × 10^5^	ZnL
Cyanidin	6	7.76 × 10^6^	ZnL
D512	6	7.85 × 10^5^	ZnL^2+^
D607 (bipyridyl-D2R/D3R agonist hybrid)	6.4	6.31 × 10^3^	ZnL
DA-2 (8D)	7.5	2.80 × 10^2^	ZnL
DA-3	6	7.85 × 10^5^	ZnL
DA-4	6	7.85 × 10^5^	ZnL
Dabigatran etexilate	6	7.85 × 10^5^	ZnL
Dabrafenib	6	8.23 × 10^7^	ZnL
(*S*)-3,4-DCPG	6	6.84 × 10^6^	ZnL
Deferricoprogen	8.3	4.35 × 10^1^	ZnHL
Delphinidin	6	2.71 × 10^5^	ZnL
Demethoxycurcumin	6	5.76 × 10^5^	ZnL
Dendropanax morbifera active compound	6	7.76 × 10^6^	ZnL
Desferrioxamine (Deferoxamine, Desferal, DFO)	7.4	3.97 × 10^2^	ZnH_2_L^+^
(*S*)-*N*-(3-(3,6-Dibromo-9*H*-carbazol-9-yl)-2-fluoropropyl) -6-methoxypyridin-2-amine	6	1.95 × 10^6^	ZnHL
4,5-*O*-Dicaffeoyl-1-*O*-(malic acid methyl ester)-quinic acid derivatives (R_1_, R_2_, R_3_, R_4_, or R_5_ = caffeoyl)	6	7.76 × 10^6^	ZnL
Dihydromyricetin	6	2.71 × 10^5^	ZnL
5-(3,4-Dihydroxybenzylidene)-2,2-dimethyl-1,3-dioxane-4,6-dione	6	7.76 × 10^6^	ZnL
7,8-Dihydroxycoumarin derivative DHC12	6	7.76 × 10^6^	ZnL
3′,4′-Dihydroxyflavone	6	7.76 × 10^6^	ZnL
7,8-Dihydroxyflavone	6	7.76 × 10^6^	ZnL
(*E*)-3,4-Dihydroxystyryl aralkyl sulfones	6	7.76 × 10^6^	ZnL
(*E*)-3,4-Dihydroxystyryl aralkyl sulfoxides	6	7.76 × 10^6^	ZnL
2-[[(1,1-Dimethylethyl) oxidoimino]-methyl]-3,5,6-trimethylpyrazine	6	2.20 × 10^6^	ZnL
DKP	6	1.52 × 10^6^	ZnL
l-DOPA (levodopa, CVT-301)	6	2.96 × 10^6^	ZnHL
DOPA-derived peptido-mimetics (deprotected)	6	2.96 × 10^6^	ZnHL
DOPA-derived peptido-mimetics (protected)	6	7.76 × 10^6^	ZnL
l-dopa deuterated	6	2.96 × 10^6^	ZnHL
Doxycycline	6.1	3.88 × 10^4^	ZnL
Droxidopa	10.9	1.18 × 10^−1^	ZnHL
Echinacoside	6	7.76 × 10^6^	ZnL
Ellagic acid	6	7.76 × 10^6^	ZnL
Entacapone (comtan, ASI-6)	6.2	2.62 × 10^4^	ZnL
Enzastaurin	6	7.85 × 10^5^	ZnL
Epicatechin	6	7.76 × 10^6^	ZnL
Epigallocatechin-3-gallate	6	1.81 × 10^13^	Zn_2_L
Etidronate (HEDPA)	7.4	4.32 × 10^2^	ZnL^2−^
Exifone	6	1.81 × 10^13^	Zn_2_L
F13714, F15599	6	2.28 × 10^5^	ZnL
Fisetin (3,3′,4′,7-tetra-hydroxy-flavone)	6	7.76 × 10^6^	ZnL
Fraxetin	6	7.76 × 10^6^	ZnL
Gallic acid derivatives	6	1.81 × 10^13^	Zn_2_L
Gallocatechin	6	2.71 × 10^5^	ZnL
Garcinol	6	7.76 × 10^6^	ZnL
Glutamine	6	8.61 × 10^5^	ZnL^+^
Glutathione derivatives	14.8	1.20 × 10^−5^	ZnH_−2_L_2_^2−^
Glutathione-hydroxy-quinoline compound	7.8	1.37 × 10^2^	ZnH_−1_L^+^
Glutathione-l-DOPA compound	6.3	1.20 × 10^4^	ZnH_−1_L
Gly-N-C-DOPA	6	2.96 × 10^6^	ZnHL
GSK2795039	9.7	1.64	ZnL_2_
Guanabenz	6	7.85 × 10^5^	ZnL
Hinokitiol	6.2	2.06 × 10^4^	ZnL^+^
8-HQ-MC-5 (VK-28)	7.5	3.16 × 10^2^	ZnL
4-Hydroxyisophthalic acid	6	9.02 × 10^7^	ZnL
1-Hydroxy-2-pyridinone derivatives	6.3	1.01 × 10^4^	ZnL
3-Hydroxy-4(1*H*)pyridinone (Deferiprone)	6.2	1.45 × 10^4^	ZnL^+^
3-Hydroxy-4(1*H*)pyridinone derivatives (R = H)	6.2	1.45 × 10^4^	ZnL
8-hydroxyquinoline-2-carboxaldehyde isonicotinoyl hydrazone	7.5	3.16 × 10^2^	ZnL
Hydroxy-quinoline-propargyl hybrids (HLA20)	7.5	3.16 × 10^2^	ZnL
Hydroxytyrosol butyrate	6	7.76 × 10^6^	ZnL
l-(7-Imino-3-propyl-2,3-dihydrothiazolo [4,5-*d*] pyrimidin-6(7*H*)-yl)urea	6.5	5.10 × 10^3^	ZnH_−1_L
Imipramine	6	1.95 × 10^6^	ZnHL^3+^
Isobavachalcone	6	2.12 × 10^5^	ZnL^+^
Isochlorogenic acid	6	6.86 × 10^5^	ZnL^−^
Isoquercetin (isoquercitrin)	6	3.94 × 10^24^	ZnH_4_L
Kaempferol	6	3.94 × 10^24^	ZnH_4_L^+^
Kaempferol, 3-*O*-a-L arabino-furanoside-7-*O*-a- L-rhamno-pyranoside	6	3.56 × 10^20^	ZnH_3_L
KR33493	6	1.54 × 10^6^	ZnL
Kukoamine	6	7.76 × 10^6^	ZnL
Lestaurtinib	6	7.85 × 10^5^	ZnL
Lipoic acid	6	3.84 × 10^6^	ZnL^+^
LY354740	6	1.46 × 10^7^	ZnL
M10 M30 (VAR10303) M99	7.5	3.16 × 10^2^	ZnL
Macranthoin G	6	7.76 × 10^6^	ZnL
Magnesium lithospermate B	6	7.76 × 10^6^	ZnL
α-mangostin	6	7.76 × 10^6^	ZnL
γ- mangostin	6	7.76 × 10^6^	ZnL
MAOI-1	6	2.71 × 10^5^	ZnL
MAOI-2	6	7.85 × 10^5^	ZnL
MAOI-4	6	5.88 × 10^5^	ZnHL
Metformin (Met)	6	1.87 × 10^6^	ZnL^+^
Methoxy-6-acetyl-7-methylijuglone	6.1	4.82 × 10^4^	ZnL_2_
*N*′-(4-methylbenzylidene)-5-phenylisoxazole-3-carbohydrazide	6	1.76 × 10^6^	ZnL
Minocycline	6.4	7.52 × 10^3^	ZnHL
Mitomycin C	6	3.49 × 10^6^	ZnL
MitoQ	7.1	8.74 × 10^2^	ZnL
Morin	6	5.69 × 10^16^	ZnH_3_L
[18F]MPPF	6	7.85 × 10^5^	ZnL
MSX-3	6	8.19 × 10^6^	ZnL
Nicotinamide adenine dinucleotide phosphate (NADPH)	6.1	3.72 × 10^4^	ZnL
Nicotinamide mononucleotide	6	8.19 × 10^6^	ZnL
Nitecapone	6.2	2.62 × 10^4^	ZnL
Nordihydroguaiaretic acid	6	7.76 × 10^6^	ZnL
Oleuropein	6	7.76 × 10^6^	ZnL
Opicapone	6.2	2.62 × 10^4^	ZnL
P7C3	6	1.96 × 10^7^	ZnL^2+^
PBF-509	6	7.85 × 10^5^	ZnL
PBT2	7.5	3.16 × 10^2^	ZnL
PBT434	7.9	1.19 × 10^2^	ZnL_2_
Petunidin	6	7.76 × 10^6^	ZnL
Phenothiazine 2Bc (n=0)	6	7.85 × 10^5^	ZnL^2+^
Phenothiazine 2Bc (n=1)	6	1.95 × 10^6^	ZnHL^3+^
Phenylhydroxamates	6	2.07 × 10^5^	ZnL
Piceatannol	6	7.76 × 10^6^	ZnL
Piperazine-8-OH-quinolone hybrid	7.5	3.16 × 10^2^	ZnL
Preladenant	6	7.85 × 10^5^	ZnL
Promethazine	6	7.85 × 10^5^	ZnL^2+^
Protocatechuic acid	6	1.25 × 10^7^	ZnL^−^
Protosappanin A	6	7.76 × 10^6^	ZnL
Pyrazolobenzothiazine-based carbothioamides	6	4.73 × 10^7^	ZnL
Pyrimidinone 8	6	7.85 × 10^5^	ZnL
Q1 Q4	7.5	3.16 × 10^2^	ZnL
Quercetin	6	3.94 × 10^24^	ZnH_4_L^+^
Quinoline derivatives as SUMOylation activators	6	3.94 × 10^6^	ZnL^2+^
Radotinib	6.4	6.31 × 10^3^	ZnL
Riboflavin	6	2.16 × 10^5^	ZnHL^3+^
Rifampicin (ASI-3)	6.1	7.04 × 10^4^	ZnL
Rimonabant	6	1.52 × 10^6^	ZnL
Rosmarinic acid	6	7.76 × 10^6^	ZnL
Rutin	6	3.94 × 10^24^	ZnH_4_L^+^
Salicylate, sodium salt	6	9.02 × 10^7^	ZnL
Salvianolic acid B	6	7.76 × 10^6^	ZnL
SCH58261 SCH412348	6	7.85 × 10^5^	ZnL
ST1535 ST4206	6	7.85 × 10^5^	ZnL
Staurosporine	6	7.85 × 10^5^	ZnL
Stemazole	6	4.73 × 10^7^	ZnL
Sulfuretin	6	7.76 × 10^6^	ZnL
Tannic acid	6	1.22 × 10^5^	ZnL
Tanshinol	6	7.76 × 10^6^	ZnL
Taurine	6	1.26 × 10^12^	ZnL_2_
Tetracycline	6	7.01 × 10^6^	ZnL
Tolcapone (ASI-7)	6.2	2.62 × 10^4^	ZnL
Tozadenant	8.7	1.62 × 10^1^	ZnL_2_
Transilitin	6.1	8.21 × 10^4^	ZnL
o-Trensox	21.7	1.95 × 10^−12^	ZnL^4−^
2′, 3′, 4′-Trihydroxyflavone	6	2.71 × 10^5^	ZnL
2,3,3-Trisphosphonate	12.1	7.93 × 10^−3^	ZnL
V81444	6	2.28 × 10^5^	ZnL
VAS3947 VAS2870	6.5	5.10 × 10^3^	ZnH_−1_L
Verbascoside	6	9.29 × 10^5^	ZnH_−1_L
WIN 55,212-2	6	7.85 × 10^5^	ZnL^2+^
WR-1065	6	1.95 × 10^6^	ZnHL^3+^
Zonisamide	8	8.81 × 10^1^	ZnL

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
