# Peer review of "Metal Chelation Therapy and Parkinson’s Disease: A Critical Review on the Thermodynamics of Complex Formation between Relevant Metal Ions and Promising or Established Drugs"

_biomolecules, 2019, doi:10.3390/biom9070269_

Reviewer 1 Report

There is a conceptual issue that needs a more critical assessment. High binding affinity for a metal is a desirable feature to ensure specificity for any chelator and stability of the chelate. In the in vivo clinical context it is not sufficient, definitely when dealing with iron in normal settings, let alone with elderly patients who are on the border or already with iron deficiency. I think that conservative chelation-as presented-is a bit limited, and been such a crucial point for justifying systemic chelation for PD, it should be better more emphasized and wider described.

Author Response

The authors thank again the reviewer for his/her further revision work. The answer to reviewer's suggestion is reported.

There is a conceptual issue that needs a more critical assessment. High binding affinity for a metal is a desirable feature to ensure specificity for any chelator and stability of the chelate. In the in vivo clinical context it is not sufficient, definitely when dealing with iron in normal settings, let alone with elderly patients who are on the border or already with iron deficiency. I think that conservative chelation-as presented-is a bit limited, and been such a crucial point for justifying systemic chelation for PD, it should be better more emphasized and wider described...

We recognised that the conservative chelation strategy should be better emphasised in our paper. To this aim we partially rewrote section 3 of our paper, and new sentences were added in this section, in section 6, and in section 7 (conclusions) of the main text. These new sentences were highlighted in yellow.

Reviewer 2 Report

Dear Authors

Thank you for considering my suggestions.  I feel the manuscript have been improved and will prove to be an important contribution to the field. So that I can get the last word, I did notice that the title on the supplement isn’t the same as that of the  title of the manuscript.

Kind Regards

Author Response

The authors thank again the reviewer for his/her further revision work. The answer to reviewer's suggestion is reported.

Dear Authors

Thank you for considering my suggestions. I feel the manuscript have been improved and will prove to be an important contribution to the field. So that I can get the last word, I did notice that the title on the supplement isn’t the same as that of the title of the manuscript.

Kind Regards

We changed the title of the Supporting Information, and now the two titles are identical.